# Physics-informed Temporal Difference Metric Learning for Robot Motion Planning

**Ruiqi Ni[1], Zherong Pan[2], Ahmed H. Qureshi[1]***
[1]Purdue University, [2]Lightspeed Studios
*ahqureshi@purdue.edu

## Abstract

The motion planning problem involves finding a collision-free path from a robot's starting to its target configuration. Recently, self-supervised learning methods have emerged to tackle motion planning problems without requiring expensive expert demonstrations. They solve the Eikonal equation for training neural networks and lead to efficient solutions. However, these methods struggle in complex environments because they fail to maintain key properties of the Eikonal equation, such as optimal value functions and geodesic distances. To overcome these limitations, we propose a novel self-supervised temporal difference metric learning approach that solves the Eikonal equation more accurately and enhances performance in solving complex and unseen planning tasks. Our method enforces Bellman's principle of optimality over finite regions, using temporal difference learning to avoid spurious local minima while incorporating metric learning to preserve the Eikonal equation's essential geodesic properties. We demonstrate that our approach significantly outperforms existing self-supervised learning methods in handling complex environments and generalizing to unseen environments, with robot configurations ranging from 2 to 12 degrees of freedom (DOF). The implementation code repository is available at `https://github.com/ruiqini/ntrl-demo`.

## 1 Introduction

Robot motion planning is a well-established problem focused on finding a collision-free path between a robot's initial and target configurations. In recent years, learning-based methods have emerged as promising tools using function approximators to generate paths at runtime efficiently. These methods, capable of handling high-dimensional configuration spaces (C-space), are typically categorized into supervised (Ichter et al., 2018; Kumar et al., 2019; Qureshi & Yip, 2018; Ichter & Pavone, 2019; Qureshi et al., 2019; 2020; Huh et al., 2021; Li et al., 2022; Fishman et al., 2023) and self-supervised (Ni & Qureshi, 2023a;b; 2024; Yang et al., 2023) learning approaches.

Supervised learning methods rely on expert demonstration trajectories for training. Expert data is often generated by classical planning algorithms, which are slow and inefficient in high-dimensional, cluttered C-spaces. As a result, these methods can have limited scalability to the cost and complexity of data acquisition (Karaman & Frazzoli, 2011; Janson et al., 2015; Gammell et al., 2015). In contrast, self-supervised learning methods eliminate the need for expert data. However, many of these approaches focus more on short-horizon local planning and workspace navigation problems (Wijmans et al., 2019; Yang et al., 2023). Recently, innovative self-supervised learning approaches, Neural Time Fields (NTFields) (Ni & Qureshi, 2023a;b), have been proposed to solve the Eikonal equation for planning problems. The Eikonal equation solution indicates the shortest travel time and defines the minimum path between two locations (Sethian, 1996). These methods avoid the need for costly expert demonstrations while offering scalable and data-efficient alternatives.

While NTFields offer promising self-supervised planning solutions, they encounter significant challenges when scaling to complex, cluttered environments and generalizing to new, unseen scenarios. This is due to their inability to fully preserve the critical properties of the Eikonal equation's solution, which should simultaneously function as an optimal value function and a geodesic distance representing the cost-to-go from start to goal configurations in the C-space.

To address the limitations of prior works, we present a novel approach that efficiently solves the Eikonal equation through temporal difference metric learning, enabling accurate solutions for the Eikonal equation in complex, cluttered environments. Our key contributions are as follows:

- We introduce temporal difference (TD) learning into physics-informed methods to solve the Eikonal equation as an optimal value function, enforcing Bellman's principle of optimality over a finite time step. By integrating TD loss with the Eikonal loss, we significantly enhance training convergence, resulting in more accurate solutions.

- We propose a novel network architecture for metric learning over the solution space of the Eikonal equation, preserving the fundamental properties of geodesic distance. Our approach ensures that the learned metric adheres to essential characteristics such as triangle inequality, symmetry, and non-negativity. By parameterizing the network within a metric space, the learning process is constrained to adhere to these properties, which improves training efficiency and stability.

- During runtime inference, we utilize a sampling-based Model Predictive Control (MPC) (Williams et al., 2016; Bharadhwaj et al., 2020) to minimize the learned value function. This approach eliminates the need for gradient computation, improving runtime efficiency. Additionally, the inherent randomness in sampling helps the system escape poorly learned regions and supports multimodal solutions, enhancing robustness in complex environments.

We evaluate our framework on complex planning tasks with C-space ranging from 2-12 DOF and demonstrate its scalability to complex scenes and generalization ability to multiple and unseen environments. Our results show that our proposed approach significantly outperformed prior state-of-the-art learning-based planning methods. Additionally, we compare our proposed metric learning approach with other metrics commonly used in Reinforcement Learning (RL) for the value function learning. Our results demonstrate that our metric better captures the key properties of the Eikonal equation, leading to a more accurate approximation of its solution.

## 2 RELATED WORK

The prior work in robot motion planning can be categorized into classical sampling or search-based approaches, trajectory optimization (TO), and learning-based methods. The classical approaches (Sethian, 1996; Karaman & Frazzoli, 2011; Janson et al., 2015; Gammell et al., 2015) suffer from poor computational efficiency in high dimensional problems as they rely on C-space discretization or collision-free sampling to find a path. The optimization-based techniques (Ratliff et al., 2009; Kalakrishnan et al., 2011) convert hard constraints into soft constraints and solve them via optimization, leading to robot paths that are often prone to local minima. In view of these pitfalls, supervised and self-supervised learning-based methods have emerged as promising alternatives that learn function generators and quickly infer robot paths at run times.

Supervised learning-based approaches learn by imitating expert demonstration data (Ichter & Pavone, 2019; Yonetani et al., 2021; Bency et al., 2019; Chaplot et al., 2021; Saha et al., 2024; Zang et al., 2023; Huh et al., 2021; Qureshi et al., 2019; 2020; Fishman et al., 2023; Ichter et al., 2018). They can learn a path generator (Qureshi et al., 2019; 2020; Fishman et al., 2023), C-space sampler for sampling-based methods (Qureshi & Yip, 2018; Ichter et al., 2018), or prior distribution for optimization-based techniques (Saha et al., 2024). Although these learning-based methods infer robot trajectories orders of magnitude faster than classical techniques, they are bottlenecked by their need for expert demonstration data, which is gathered by running computationally intensive classical planners at a large scale.

Unlike expert-driven methods, self-supervised learning eliminates the need for labeled data. Reinforcement learning (RL) approaches (Sutton, 2018) learn through trial-and-error interactions but struggle with sparse rewards, often relying on expert demonstrations (Vecerik et al., 2017). More recently, physics-informed neural motion planners (Ni & Qureshi, 2023a;b; 2024; Li et al.; Liu et al., 2024; Shen et al., 2024) have emerged, solving the Eikonal equation for motion planning by minimizing equation loss on offline-sampled points. This can be viewed as offline goal-conditioned RL. A key yet overlooked aspect of these methods is that the Eikonal equation defines both an optimal value function and a geodesic distance on a Riemannian manifold. Failing to capture these properties hinders scalability in complex, cluttered, and unseen environments.

Our approach frames the travel time field as a metric space, drawing from Quasimetric Reinforcement Learning (QRL) (Schaul et al., 2015; Zhang et al., 2020; Bellemare et al., 2019; Wang et al., 2023), which enforces properties like non-negativity and the triangle inequality. However, existing QRL methods focus on short-horizon tasks and overlook collision avoidance in complex settings. In contrast, our method scales to long-horizon tasks in high-dimensional C-spaces while integrating complex collision constraints. We further show that Eikonal equation solutions form true metrics, supporting multiple shortest paths in multi-connected regions, and introduce novel metric functions to capture these properties.

## 3 BACKGROUND

In this section, we formalize the general robot motion planning problem and then review the physics-informed neural motion planner proposed by (Ni & Qureshi, 2023a;b), which is the basis of our method.

### 3.1 ROBOT MOTION PLANNING

We denote by $\mathcal{X} \subset \mathbb{R}^m$ the robot's workspace, where $m \in \mathbb{N}$ corresponds to the space's physical dimensions. The C-space is represented as $\mathcal{Q} \subset \mathbb{R}^d$, where $d \in \mathbb{N}$ reflects the robot's degrees of freedom. The workspace consists of both the obstacle-occupied region, denoted by $\mathcal{X}_{obs} \subset \mathcal{X}$, and the obstacle-free region, denoted as $\mathcal{X}_{free} = \mathcal{X} \setminus \mathcal{X}_{obs}$. We assume environment obstacles $\mathcal{X}_{obs}$ are known a prior. In the C-space, the obstacle and obstacle-free regions are given by $\mathcal{Q}_{obs} \subset \mathcal{Q}$ and $\mathcal{Q}_{free} = \mathcal{Q} \setminus \mathcal{Q}_{obs}$, respectively. The motion planning problem involves finding a trajectory that connects a given start point $q_s$ to a goal point $q_g$, such that the entire trajectory lies within $\mathcal{Q}_{free}$.

### 3.2 PHYSICS-INFORMED NEURAL MOTION PLANNER

Our approach is based on the recently proposed NTFields (Ni & Qureshi, 2023a). This method builds on the theory of the Eikonal equation, which models the wave propagation from a start point $q_s$ to the entire obstacle-free C-space. The solution of the Eikonal equation can be represented by the time for the wavefront starting from $q_s$ to reach $q_g$, with the shortest path being traced by following the negative gradient of the travel time (Sethian, 1996). This time field is represented as a learnable travel-time function $T(q_s, q_g)$ from $q_s$ to $q_g$. Specifically, to ensure that the travel-time is always positive and symmetric, i.e., $T(q_s, q_g) = T(q_g, q_s)$, NTFields parameterize the travel-time function by distorting the Euclidean distance as follows: $T(q_s, q_g) = \|q_s - q_g\|/\tau(q_s, q_g)$, with $\tau(q_s, q_g)$ being the distance distortion function parameterized using a neural network. They also show that the travel speed of the wavefront is inversely proportional to the gradient norm of travel time, i.e.:

$$1/S(q_g) = \|\nabla_{q_g} T(q_s, q_g)\| \quad 1/S(q_s) = \|\nabla_{q_s} T(q_s, q_g)\|, \tag{1}$$

where $S(q)$ is the wavefront's travel speed. To ensure that the robot moves in the free space $\mathcal{Q}_{free}$, and stops in the obstacle space $\mathcal{Q}_{obs}$, NTFields introduced a ground truth speed $S^\star(q)$ based on the truncated distance between $q$ and $\mathcal{X}_{obs}$ as follows:

$$S^\star(q) = \mathrm{clip}(\mathrm{d}_{obs}(q, \mathcal{X}_{obs})/d_{max}, d_{min}/d_{max}, 1). \tag{2}$$

Here, $\mathrm{d}_{obs}(q, \mathcal{X}_{obs})$ represents the minimal distance between workspace robot geometry at configuration $q$, computed using forward kinematics, and the workspace obstacles $\mathcal{X}_{obs}$. We further assume $\mathrm{d}_{obs}$ is differentiable in $q$ using differentiable forward kinematics (Villegas et al., 2018). The parameters $d_{min}$, and $d_{max}$ are the minimum and maximum distance thresholds, respectively. NTFields are then trained by minimizing the following Eikonal loss between the ground truth speed $S^\star(q)$ and the speed $S(q)$ predicted by the neural network parametrized $T(q_s, q_g)$ and Eq. 1:

$$L_E = (\sqrt{S^\star(q_s)/S(q_s)} - 1)^2 + (\sqrt{S^\star(q_g)/S(q_g)} - 1)^2. \tag{3}$$

Intuitively, the Eikonal equation seeks to approximate the optimal value function $T(q_s, q_g)$ with the desirable gradient norm, and the gradient of value function gives the desirable motion direction during inference. The subsequent work, P-NTFields (Ni & Qureshi, 2023b) introduced curriculum learning and viscosity Eikonal equation to enhance the training process. However, local minima persist, leading to incorrect solutions that hinder success in complex environments. More recently, PC-Planner (Shen et al., 2024) incorporated monotonicity and optimality constraints to refine results further. However, it does not explicitly leverage the Eikonal equation's role as an optimal value function and geodesic distance, which are crucial for global consistency and optimal path planning.

## 4 METHODS

In this section, we present our novel approach that efficiently approximates the solution of the Eikonal equation via temporal difference metric learning for robot motion planning in complex, cluttered environments. We observe two key properties of the Eikonal equation that can be utilized to enhance the effectiveness of self-supervised learning. First, we show that the solution of the Eikonal equation can be interpreted as the optimal value function in an optimal control problem. Therefore, the Bellman optimality principle can be applied both at infinitesimal and finite time scales. While prior works only apply the infinitesimal perspective to formulate their loss function, we propose that combining the infinitesimal and finite time-scale perspectives can effectively avoid local overfitting (Sec. 4.1). Second, we highlight that the travel-time function can be understood as the geodesic distance on a Riemannian manifold. As a result, we propose a novel network architecture compatible with generalizable metric learning (Sec. 4.2).

### 4.1 THE OPTIMAL CONTROL PERSPECTIVE

We highlight that the solution of the Eikonal equation and that of an optimal control problem coincide. Let us consider a fully actuated $d$-DOF robot with configuration $q(t)$ and control signal $u(t)$ at time instant $t$. The robot is governed by the trivial dynamics, $\dot{q} = u$, such that $\|u(t)\| = 1$. From $q_s$ to $q_g$, we have by the classical optimal control theory that the optimal travel time satisfies:

$$T(q_s, q_g) = \min_{u(t)} \int_{t_s}^{t_g} \|\dot{q}(t)\| / S^\star(q(t)) dt \tag{4}$$
$$\text{s.t.} \quad q(t_s) = q_s, \quad q(t_g) = q_g, \quad \|u(t)\| = 1.$$

Note that the above formulation inherently avoids obstacles due to the definition of $S^\star(q(t))$ tending to zero as the configuration tends to $\mathcal{Q}_{obs}$. In our Appendix. A, we show that the optimal solution of Eq. 4 under optimal actions, $u_s^\star = -\nabla_{q_s} T(q_s, q_g) / \|\nabla_{q_s} T(q_s, q_g)\|$, also satisfies Eq. 1 by an infinitesimal perturbation analysis. Such analysis also shows that the Eikonal equation captures the Bellman optimality principle at an infinitesimal time scale.

#### 4.1.1 TEMPORAL DIFFERENCE LOSS

We observe that the value function trained using only $L_E$ (Eq. 3) struggles to capture the broader, globally optimal structures of the value function, limiting its performance in more complex environments. This is because relying solely on loss functions, such as $L_E$, at an infinitesimal time scale can cause overfitting problems as they only regulate the gradient of the network, acting as a tangent matching term (Simard et al., 2002). However, since our network only takes a sparse set of samples to evaluate $L_E$, the tangent function's landscape between these samples is uncontrolled and can significantly deviate from the ground truth. Furthermore, such local, erroneous tangent landscapes will be reflected as a global error in the value function's landscapes.

To address the above-mentioned issues, we introduce the discrete Bellman loss, resembling the TD loss widely used in RL (Sutton, 2018) to complement the Eikonal loss $L_E$. TD learning approximates the value function by considering the differences between successive state values over a small but finite time step, effectively contrasting the value functions at two nearby spots. Specifically, our TD loss $L_{TD}$ is derived using a Taylor expansion of the value function along the optimal policy $u^\star$ with a small time step $\Delta t$:

$$L_{TD} = \left[ T(q_s, q_g) - \Delta t / S^\star(q_g) - T(q_s, q_g + u_g^\star \Delta t) \right]^2 + \left[ T(q_s, q_g) - \Delta t / S^\star(q_s) - T(q_s + u_s^\star \Delta t, q_g) \right]^2, \tag{5}$$

where we define $u_g^\star = -\nabla_{q_g} T(q_s, q_g) / \|\nabla_{q_g} T(q_s, q_g)\|$ by its symmetry with $u_s^\star$. Please refer to Appendix. A for a derivation of our TD loss

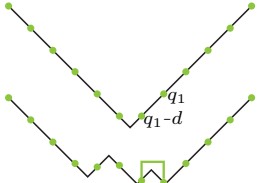

Figure 1: The plots depict the solution to the Eikonal equation $|\nabla_q T(q)| = 1, T(0) = 0$ with sampled green points as training data. In the top plot, the desired solution $T(q) = |q|$ is shown. In contrast, the bottom plot demonstrates that the green points also satisfy $|\nabla_q T(q)| = 1$, but without proper constraints, the solution deviates. The TD loss $L_{TD}$ ensures correctness by enforcing, for example, that for $q_1 > 0$, $T(q_1) - T(q_1 - d) = d$. Therefore, $L_{TD}$ is more capable of finding the ground truth solution.

function. The TD loss can be understood as an upwind scheme for the value function, ensuring that the value of a given state equals the local cost plus the value of the next state after following the optimal policy. Essentially, it ensures proper value propagation, maintaining consistency with the optimal policy. In Fig. 1, we show that a suboptimal solution can have a zero $L_E$ on sampled points but a large $L_{TD}$. TD loss also serves as a finite difference approximation of the Eikonal loss $L_E$. Thus two loss terms, $L_E$ and $L_{TD}$, are complementary. The TD loss captures the optimal substructure over a finite region—providing a coarse-grained view of value propagation—the Eikonal loss focuses on the optimal substructure at an infinitesimal scale, offering a fine-grained, continuous perspective. By combining these two losses, we achieve both global and local consistency in the value function's landscapes, leading to more accurate results.

### 4.1.2 OBSTACLE-AWARE NORMAL ALIGNMENT

Our method of training value function is an instance of differential dynamic programming, where the correct value function propagates from start to goal. Therefore, when the network makes a local mistake in the value function, such propagation can exaggerate the consequence. We find such an effect particularly detrimental at an early stage of training, especially when the network predicts a non-zero velocity inside the obstacle $\mathcal{X}_{obs}$. In this case, a shortcut is created through the obstacle, and the downstream value functions are significantly underestimated. Fortunately, we know how the policy should behave when the robot is near obstacles. In such cases, the desired velocity $u^\star$ should naturally align with the normal direction of the obstacle's surface to avoid collisions. As the robot moves farther away from obstacles, this alignment becomes less relevant, and the influence of the normal direction diminishes. Based on such observation, we introduce the following normal alignment loss:

$$L_N = (1 - S^\star(q_s))\|S^\star(q_s)\nabla_{q_s}T(q_s, q_g) + \nabla_{q_s}S^\star(q_s)/\|\nabla_{q_s}S^\star(q_s)\|\|^2 + \\ (1 - S^\star(q_g))\|S^\star(q_g)\nabla_{q_g}T(q_s, q_g) + \nabla_{q_g}S^\star(q_g)/\|\nabla_{q_g}S^\star(q_g)\|\|^2, \tag{6}$$

where $\nabla_{q_s}S^\star(q_s)/\|\nabla_{q_s}S^\star(q_s)\|$ computes the normal direction in C-space, and the weight term $1 - S^\star(q_s)$ ensures the loss only takes effect near the obstacles. Note that evaluating $L_N$ relies on the differentiable assumption of the function $d_{obs}$ discussed in Sec. 3.2.

### 4.1.3 CAUSALITY PRESERVATION

The importance of causality has been noticed since the invention of the fast sweeping method (Van Trier & Symes, 1991) for solving the Eikonal equation. Specifically, the TD loss should propagate the value in the ascending direction of travel time but not the other way around. Although TD loss $L_{TD}$ encourages this propagation, the random batch training in neural networks does not maintain the required one-way information flow. To avoid this issue and preserve causality when using neural approximation, we use the following causality weight recently introduced by (Wang et al., 2024b):

$$L_C = \exp(-\lambda_C T(q_s, q_g)) \tag{7}$$

$L_C$ encourages the optimizer to prioritize learning smaller values associated with closer-to-start states before tackling larger value functions associated with more distant states. By gradually learning in this manner, the model follows the desirable unilateral information-propagation direction, which ensures smoother convergence. Put together, our final loss function is formulated as:

$$L = (\lambda_E L_E + \lambda_{TD} L_{TD} + \lambda_N L_N)L_C, \tag{8}$$

where $\lambda_E, \lambda_{TD}, \lambda_N, \lambda_C$ are hyper-parameters that control the contribution of the respective losses. Please refer to Appendix. C for details on the choice of these hyperparameters.

To summarize, we enhance the value function's convergence in three ways. First, we enforce Bellman's principle of optimality under infinitesimal and finite time scales using $L_E$ and $L_{TD}$, respectively. Further, our normal alignment loss $L_N$ mitigates policy misalignment near obstacles, providing a strong prior at an early stage of training. Finally, the causality weight $L_C$ ensures a natural direction of information propagation in learning, prioritizing smaller values first. These contributions ensure the performance of learned value functions under complex environments.

## 4.2 GENERALIZABLE METRIC LEARNING

In the previous section, we demonstrated that a properly designed loss function can train more accurate value functions. In this section, we show that performance can be further enhanced by employing a carefully designed network architecture that constrains the solution within a metric space. Specifically, we introduce a network architecture compatible with metric learning and an attention mechanism that allows our method to generalize to unseen environments.

### 4.2.1 METRIC LEARNING

We notice that the travel-time function $T(q_s, q_g)$ can be understood as a geodesic distance on the Riemannian manifold with a metric of $I/S^\star(q)$. Under such a setting, the optimal curve $q(t)$ represents the geodesic curve on the manifold, with the corresponding geodesic distance given in Eq. 4. Clearly, the metric $T(q_s, q_g)$ should satisfy the three fundamental properties: **(Non-negativity)** For $q_s \neq q_g$, $T(q_s, q_g) > 0$; and for $q_s = q_g$, $T(q_s, q_g) = 0$; **(Symmetry)** $T(q_s, q_g) = T(q_g, q_s)$; **(Triangle inequality)** For any intermediate point $q_m$, $T(q_s, q_g) \leq T(q_s, q_m) + T(q_m, q_g)$. While previous works (Ni & Qureshi, 2023a;b) have preserved the properties of non-negativity and symmetry, they often break the triangle inequality by using the factorized form $T(q_s, q_g) = \|q_s - q_g\|/\tau(q_s, q_g)$. This allows for situations where an indirect path between two points may be computed as shorter than the direct path. Instead, we employ metric learning using a network $f_\theta(\cdot)$ to map the start and goal to a high-dimensional latent space and then define:

$$T(q_s, q_g) = D(f_\theta(q_s), f_\theta(q_g)), \tag{9}$$

with $D(\cdot, \cdot)$ being some metric function satisfying the three aforementioned properties.

The definition of metric function $D$ is crucial to the performance of our method. Due to the invariance of metric properties under transformation $f_\theta$, our travel-time function will also retain those desirable properties. Several prior works used the Euclidean distance (Carroll & Arabie, 1998; Panozzo et al., 2013). However, the geodesic distance in low-dimensional space cannot always be embedded as an Euclidean distance in high-dimensional space after nonlinear transformations according to (Pitis et al., 2020; Pang et al., 2023). Euclidean distance corresponds to a single shortest path, while geodesic distance can represent multiple shortest paths in multi-connected regions. To accommodate this property, we propose using the $L_1$ distance, which permits multiple latent paths to share the same distance. Empirically, we combine the merit of $L_\infty$ and $L_1$ distance. Specifically, we define the latent space to have a dimension of $a \times b$, i.e. $f_\theta : \mathbb{R}^d \to \mathbb{R}^{a \times b}$. Our proposed distance metric is then computed as follows:

Figure 2: The figure shows the geodesics of points $A$ and $B$ on a circle under different embeddings. With $L_2$, the distance collapses into a line, causing overlap and ambiguity. In contrast, $L_1$ transforms it into a diamond, preserving the geodesic structure and resolving ambiguity.

$$D(x, y) = \sum_{i=1}^{a} \left[ \max_{1 \leq j \leq b} |x_{i,j} - y_{i,j}| \right], \tag{10}$$

where the maximum along one dimension computes the $L_\infty$ distance, while the summation along the other dimension computes the $L_1$ distance. We further motivate our distance metirc through illustration in Fig. 2. The two half-circle paths between points $A$ and $B$ are better captured by our distance metric than Euclidean distance as it transforms the circle into a 2D diamond to preserve multipath solutions.

### 4.2.2 ATTENTION MECHANISM

Prior physics-informed neural motion planners (Ni & Qureshi, 2023a;b) can only represent the solution of the Eikonal equation for a given environment. In other words, they are unable to scale or generalize to multiple environments. In contrast, with our more accurate training techniques, we enable a generalizable neural Eikonal equation solver by conditioning our feature encoder, $f(\cdot)$, on the environment shape $\mathcal{X}_{obs}$. To this end, we assume that all the environments are represented using a point cloud, also denoted as $\mathcal{X}_{obs}$. As a result, we can use the state-of-the-art PointNext encoder (Qian et al., 2022) to compute a global latent feature $\mathbf{z} = PointNext(\mathcal{X}_{obs})$. Next, given a configuration point $q$, we first compute a fixed random positional encoding $\gamma(q) = [\sin(2\pi \mathbf{b}^T q), \cos(2\pi \mathbf{b}^T q)]$,

where $\mathbf{b}$ is a fixed random Gaussian matrix. Finally, we treat $\gamma(q)$ as the query and $\mathbf{z}$ as the keys and values and compute the conditioned feature using the attention mechanism (Vaswani, 2017; Rebain et al., 2022). The attention mechanism enables the network to selectively focus on relevant parts of the environment, dynamically adjusting its output based on the spatial context provided by the point cloud. Let the attention mechanism be denoted as $att(\cdot)$ providing conditional features as $att(\gamma(q), \mathbf{z})$, we feed the resulting attention output into the PirateNets structure $g(\cdot)$ (Wang et al., 2024a), which integrates a modified MLP (Wang et al., 2021) with the residual gate (Savarese & Figueiredo, 2017; He et al., 2016) for enhanced performance and stability. Put together, we have the conditional feature representation $f_\theta(q, \mathcal{X}_{obs}) = g(att(\gamma(q), \mathbf{z}))$. Finally, we compute the geodesic distance conditioned on point cloud $\mathcal{X}_{obs}$ as $T(q_s, q_g, \mathcal{X}_{obs}) = D(f_\theta(q_s, \mathcal{X}_{obs}), f_\theta(q_g, \mathcal{X}_{obs}))$. This formulation allows our model to approximate Eikonal solutions for unseen environments. Moreover, we train our model by minimizing the loss (Eq. 8), please refer to Appendix. C for more details.

### 4.3 Sampling-based MPC for Path Inference

After training our value function $T(q_s, q_g)$, we use it as a cost-to-go function in a sampling-based MPC (Williams et al., 2016; Bharadhwaj et al., 2020) framework for path planning. We begin by randomly sampling actions $u$ from a zero-mean normal distribution, which modifies the current configuration to $q(t + 1) = q(t) + u$. The distribution's mean is updated after each iteration to the last selected action. Each sampled action leads to the next configuration and is assigned a value from the cost-to-go function. A softmax function is then applied to favor actions with lower travel times, allowing us to calculate a weighted average of the sampled actions. We use a receding horizon approach to sample additional actions for a fixed horizon and generate multiple rollouts, selecting the trajectory with the lowest cost-to-go. This process is repeated until we find a path between the starting and goal configurations or until we hit a time limit. Unlike prior methods (Ni et al., 2021; Ni & Qureshi, 2023b) that rely on gradient descent for path inference, our approach eliminates the need for gradient computations, enhancing efficiency and allowing the system to escape local minima in the value function. We should also highlight that our cost-to-go function can also be integrated with other downstream planning approaches, such as T-RRT with completeness-guarantees (Jaillet et al., 2010), providing a more robust framework for planning.

## 5 Experiments and Analysis

This section presents our experiments and their analysis. We begin with an ablation study to demonstrate the effectiveness of our loss function and metric representation. Next, we provide scalability and generalization analysis to showcase our method's ability to scale and perform well in complex, high-dimensional, and unseen environments. To evaluate our method, we compare it against the following baselines from prior planning methods.

- **Traditional Methods:** For the search-based approach, we employ the Fast Marching Method (FMM) (Sethian, 1996), which solves the Eikonal equation but is limited to 3D environments due to the curse of dimensionality. For sampling-based planner (SMP), we use RRTConnect (RRTC) (Kuffner & LaValle, 2000) and LazyPRM (L-PRM) (Bohlin & Kavraki, 2000), followed by path smoothing, to find paths in tasks ranging from 3 to 12 degrees of freedom (DOF). These methods are probabilistically complete and we allow a maximum of 30 seconds for each algorithm to find solutions.

- **Supervised Learning:** We use MPNet (Qureshi et al., 2019) for our 3D environments. For 7D manipulation tasks, we use MPiNet (Fishman et al., 2023) instead of MPNet. Note that MPiNet is specifically designed for robot manipulators and has been shown to perform better than MPNet in these tasks. Both methods learn by imitating the data collected by the expert classical methods.

- **Self-supervised Learning:** We consider NTFields (NTF) (Ni et al., 2021) and P-NTFields (P-NTF) (Ni & Qureshi, 2023a) for all presented tasks. Since their original neural architecture cannot scale across multiple environments, we enhance them with our proposed attention-based environment encoding to evaluate their generalization and scalability. In a 2D maze, we also compare our TD learning with several metric representations considered in the RL literature.

For performance metrics, we evaluate the success rate (SR), path length, and computational times. SR shows the proportion of planning cases solved by a motion planner. Path length refers to the

total configuration-space Euclidean distance between the waypoints on the path found by a planner. Lastly, computational time represents wall clock time taken by a CPU-based execution of a planner.

## 5.1 ABLATION ANALYSIS

This section ablates our loss functions $L$ and its components, including Eikonal loss $L_E$, temporal difference loss $L_{TD}$, obstacle alignment loss $L_N$, and causality-based prioritization $L_C$. Additionally, we assess our metric formulation and compare it with other metrics commonly used in RL literature for value function learning. To conduct these evaluations, we set up complex 2D maze environments. One of the maze setups is shown in Fig. 3, and more scenarios are available in Appendix. B. We chose FMM as an expert baseline for comparison. We present the error as the mean absolute difference between the travel time of each method and the ground truth FMM, measured at grid points. However, we should highlight that these results are better understood with the illustration of contours in Fig. 3. In the figure, the contour lines show the travel time from various points to a specific point in the maze. By following the negative gradient of the travel time, a global path can be efficiently determined. Additionally, the colors represent the speed field within the maze, with yellow indicating free space and dark blue marking obstacle regions.

The **first row** in Fig. 3 presents our method, FMM, NT-Fields, and P-NTFields. NTFields and P-NTFields show significant distortion, failing to generate valid paths in cluttered environments. Our method, with a small error of 0.08 compared to 0.58 for NTFields and P-NTFields. Besides their contour lines also are incorrect, resulting in no path solution to some points in the environment.

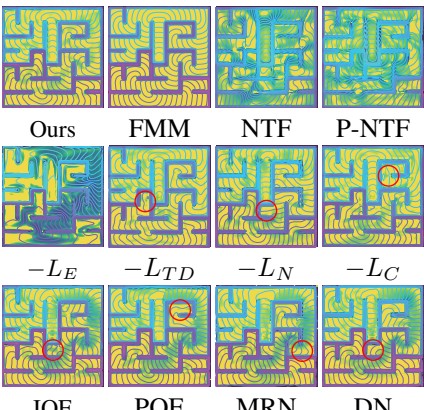

The **second row** in Fig. 3 shows the results of our method without $L_E$, $L_{TD}$, $L_N$, and $L_C$. The errors for these variations are 1.13, 0.21, 0.13, and 0.12, respectively, compared to 0.08 for our full method. The contour features show that results are incorrect with $L_E$ alone. Without $L_{TD}$, the value function develops incorrect contours at some corners of the maze. Without $L_N$ or $L_C$, our training starting from random initialization leads to incorrect convergence to local minima. These results highlight that $L_{TD}$ and $L_E$ are complementary to each other while $L_N$ and $L_C$ facilitate in correct convergence of those loss functions when starting from random initialization.

Figure 3: First row: We compare our method with FMM, NTF, and P-NTF. Middle row: We ablate our method by removing $-L_E$, $-L_{TD}$, $-L_N$, and $-L_C$. Third row: We replace our distance metric $D$ with IQE, PQE, MRN, and DN.

The **third row** in Fig. 3 presents the results of our method using alternative metric formulations commonly used in RL for value function learning. Specifically, we replace our metric (Eq. 10) with Interval Quasimetric Embeddings (IQE) (Wang & Isola, 2022a; Wang et al., 2023), Poisson Quasimetric Embeddings (PQE) (Wang & Isola, 2022b), Metric Residual Networks (MRN) (Liu et al., 2023), and Deep Norm (DN) (Pitis et al., 2020) while keeping rest of our method same as presented for fairness. The errors for IQE, PQE, MRN, and DN are 0.32, 0.46, 0.19, and 0.29, respectively, compared to 0.08 for our method. These results demonstrate that existing RL metric formulations struggle to capture the optimal value function, while our method effectively captures this through a combination of $L_\infty$ and $L_1$ distance.

In conclusion, the contour lines produced by our method align well with those from the FMM, while other methods display noticeable artifacts. The second row shows a greater emphasis on the Eikonal loss $L_E$, but it's important to note that the loss function assesses contour line accuracy across the entire C-space, whereas spurious local minima are localized issues. Consequently, improvements in loss may not seem significant despite the inclusion of additional loss terms $L_{TD}$, $L_N$, and $L_C$. Nonetheless, these differences in contour lines matter, as even a single spurious local minimum can trap the robot, preventing it from reaching the true goal configuration, as also indicated by red circles in Fig. 3. Finally, our results also highlight that standard offline RL using only TD loss is inadequate. The first plot in the second row shows that without Eikonal loss $L_E$, the method produces inaccurate results. Eikonal loss is essential and must work together with TD and other loss functions to accurately infer value functions and represent geodesic distance.

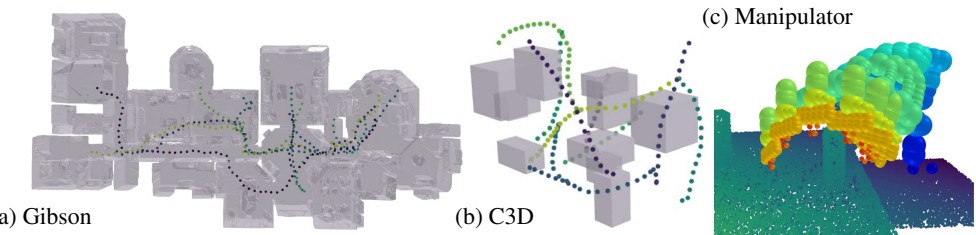

(c) Manipulator

(a) Gibson     (b) C3D

Figure 4: Depiction of our (a) Gibson, (b) Cluttered 3D (C3D), and (c) 7-DOF Manipulator environments. We also illustrate multiple trajectories planned by our method between different start and goal pairs. It can be seen that our method finds smooth trajectories while avoiding collisions.

## 5.2 SCALABILITY TO COMPLEX AND HIGH-DIMENSIONAL ENVIRONMENTS

We demonstrate our method's ability to handle complex indoor home-like environments and high-dimensional tasks.

**Indoor 3D home-like environments:** We selected ten environments from the Gibson dataset (Li et al., 2021), with room counts ranging from 7 to 16 and dimensions between 90 and 430 square meters. An example Gibson environment is depicted in Fig. 4, and more examples are available in the Appendix. B. In each environment, we evaluated 100 unseen start and goal pairs. Table 1 (a) presents the results for all methods in these environments.

Our method consistently achieves a high SR and the lowest computational planning times. While classical methods also exhibit high SR, they are significantly slower in terms of computational time. Among the learning-based methods, NTFields and P-NTFields have lower computational times but poor SR. Since our method uses MPC in contrast to travel time gradients for path inference, we have included a variant, denoted as Our-G, that incorporates gradients instead of MPC for path inference, similar to NTFields and P-NTFields. The results show that MPC improves both planning times and SR. Even with gradient-based path inference, our method outperforms NTFields and P-NTFields, indicating more accurate convergence to the Eikonal solution.

In summary, our method scales well to complex 3D home-like environments, delivering high success rates and extremely low planning times, validating its ability to solve motion planning more accurately than previous self-supervised learning methods.

| (a) Indoor Gibson | | | |
|---|---|---|---|
| Method | Time (s) | Length | SR(%) |
| Ours | $0.056 \pm 0.037$ | $5.39 \pm 3.25$ | 95.1 |
| Ours-G | $0.074 \pm 0.057$ | $7.10 \pm 5.03$ | 92.6 |
| NTF | $0.074 \pm 0.068$ | $8.12 \pm 9.45$ | 68.1 |
| P-NTF | $0.057 \pm 0.051$ | $7.13 \pm 7.43$ | 81.2 |
| FMM | $0.74 \pm 0.11$ | $4.96 \pm 2.83$ | 86.4 |
| RRTC | $2.14 \pm 1.92$ | $4.92 \pm 1.72$ | 99.7 |
| L-PRM | $0.47 \pm 0.31$ | $5.02 \pm 1.15$ | 97.8 |

| (b) Cluttered 3D (C3D) | | | |
|---|---|---|---|
| Method | Time (s) | Length | SR(%) |
| Ours | $0.025 \pm 0.011$ | $0.69 \pm 0.29$ | 99.4 |
| NTF | $0.029 \pm 0.012$ | $0.64 \pm 0.26$ | 93.8 |
| P-NTF | $0.031 \pm 0.025$ | $0.66 \pm 0.27$ | 93.4 |
| MPNet | $0.22 \pm 0.30$ | $0.68 \pm 0.31$ | 97.0 |
| FMM | $0.76 \pm 0.09$ | $0.65 \pm 0.25$ | 100 |
| RRTC | $0.16 \pm 0.13$ | $0.65 \pm 0.26$ | 100 |
| L-PRM | $0.13 \pm 0.11$ | $0.68 \pm 0.31$ | 100 |

| (c) 7-DOF Manipulator | | | |
|---|---|---|---|
| Method | Time (s) | Length | SR(%) |
| Ours | $0.074 \pm 0.029$ | $1.95 \pm 0.82$ | 87.0 |
| NTF | $0.063 \pm 0.017$ | $1.63 \pm 0.64$ | 74.3 |
| P-NTF | $0.061 \pm 0.017$ | $1.68 \pm 0.68$ | 73.3 |
| MPiNet | $0.57 \pm 0.21$ | $2.57 \pm 1.16$ | 91.0 |
| RRTC | $0.42 \pm 0.30$ | $2.04 \pm 1.06$ | 97.7 |
| L-PRM | $0.25 \pm 0.51$ | $2.01 \pm 1.04$ | 97.7 |

Table 1: Performance comparison on Gibson, C3D, and 7-DOF manipulator datasets.

**12-DOF dual-arm in real-world confined cabinet:** In this experiment, we demonstrate the ability of our method to scale to a high-dimensional 12-DOF C-space and exhibit sim2real generalization. Our real-world environment is depicted in Fig. 5. We randomly sampled 100 start and goal pairs in this environment for the testing. In this task, our method achieves a high SR of 91% with significantly low planning times of about 0.09 seconds on average. Fig. 5 depicts a demo trajectory from our method. We also include another demo trajectory in the Appendix. B. Note that these demonstrations are to exhibit the sim2real generalization of our method in high-dimensional tasks. We should also highlight that the prior self-supervised methods, i.e., NTFields and P-NTFields failed to converge in these high-DOF tasks with confined narrow passages.

## 5.3 GENERALIZATION TO NOVEL ENVIRONMENTS

This section presents the ability of our method to generalize to multiple seen and unseen environments in the following scenarios. We present the commutative results of all methods on both seen

Start                                        Intermediate                                        Goal

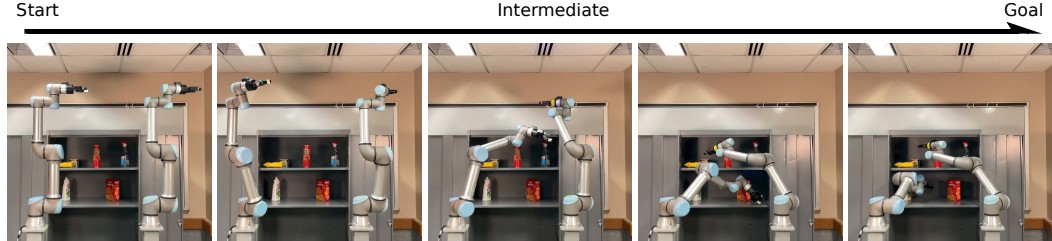

Figure 5: Demontration of a path planned by our method to navigate real-world cabinet environment using a 12-DOF dual-arm robot. This particular trajectory was planned in 0.11 seconds.

and unseen environments, while Appendix. B details performances on seen and unseen tasks separately.

**3D Cluttered Environments:** These environments are taken from the C3D dataset (Qureshi et al., 2019; 2020) and consist of 10 cubes of varying sizes randomly placed in a 3D space. An example scenario is shown in Fig. 4. For these tasks, we selected 100 seen and 100 unseen environments. The models were trained on the seen environments. For testing, we chose 500 random start and goal pairs across both seen and unseen environments. Table 1 (b) summarizes the results of all methods.

In this setting, our method achieves an SR of 99.4%, with an extremely low average planning time of 0.025 seconds. NTF and P-NTF show relatively lower SR with similar inference time. MPNet shows a similar SR but is about 10 times slower than our method and requires expert data. Classical methods, though reliable with a 100% SR, are at least 5 times slower than our approach. These results demonstrate that our method exhibits strong generalization to unseen tasks while maintaining our computational performance and a high SR.

**7-DOF Robot Arm Motion among Obstacle Clutters:** These tasks, adopted from the MPiNet dataset (Fishman et al., 2023), require a 7-DOF robot arm to navigate among multiple obstacle blocks on a tabletop. An example scene is depicted in Fig. 4. We choose 150 seen and 150 unseen environments and train neural models on the seen environment. For testing, we select 300 start and goal pairs in both seen and unseen environments. The performance is summarized in Table 1 (c). It can be viewed that our method persistently retains low computational times compared to other methods. Our SR is also considerably high, i.e., 87%, and closer to MPiNet's 91%. NTF and P-NTF show relatively lower SR with similar inference time as ours. MPiNet learned from expert data whereas our method, despite being self-supervised, exhibits high performance. The classical methods, similar to their prior experiments, demonstrate high SR but slower planning times.

## 6    CONCLUSIONS AND FUTURE WORKS

This paper introduces a new, scalable, self-supervised neural approach for solving the Eikonal equation in robot motion planning. In this paper, we highlight that the solution to the Eikonal equation can be expressed as the value function of an optimal control problem, as well as the geodesic distance of a Riemannian manifold. These perspectives have led to our novel temporal difference metric learning approach for solving the Eikonal equation more accurately. By combining this with sampling-based MPC for path inference, our method has achieved a higher success rate and lower computational cost for path inference than prior approaches. Additionally, our attention mechanism enables us to solve the Eikonal equation for unseen environments, which was previously not possible with prior self-supervised learning methods in robot motion planning.

In our future work, we aim to further enhance the generalization ability of our method. Currently, in our approach, we still observe some failure cases when generalizing to unseen environments. Specifically, our method struggles to generalize effectively to multiple, unseen environments in complex Gibson datasets. We plan to mitigate this issue by experimenting with more expressive neural encoders for the environment, such as the point transformer (Zhao et al., 2021). Lastly, we also aim to extend our approach to tackle motion planning tasks under partial observability and kinodynamic constraints.

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

## APPENDIX

This appendix provides the derivation of our TD loss, additional visualization of results, and implementation details of our method, along with the training procedure.

## A  TD LOSS DERIVATION

This section provides the derivations connecting the optimal control problem with the Eikonal equation and the TD loss. To solve the following optimal control problem:

$$T(q_s, q_g) = \min_{q(t)} \int_{t_s}^{t_g} \frac{1}{S^\star(q(t))} \|\dot{q}(t)\| dt, \quad q(t_s) = q_s,\, q(t_g) = q_g,\, \|u\| = 1, \tag{11}$$

We begin with the Taylor expansion of optimal value function $T(q_s, q_g)$ along the optimal policy direction $u_s^\star$ with a small step $\Delta t$:

$$T(q_s + u_s^\star \Delta t, q_g) = T(q_s, q_g) + \langle \nabla_{q_s} T(q_s, q_g), u_s^\star \rangle \Delta t + o(\Delta t), \tag{12}$$

where $\langle \cdot, \cdot \rangle$ denotes the inner product. Furthermore, according to Bellman's principle of optimality, with the optimal policy, the updated value function remains optimal, i.e.:

$$T(q_s + u_s^\star \Delta t, q_g) + \int_{t_s}^{t_s + \Delta t} \frac{1}{S^\star(q(t))} dt = T(q_s, q_g) + o(\Delta t). \tag{13}$$

We then compare the two equations above and tending $\Delta t \to 0$ to yield:

$$\frac{1}{S^\star(q_s)} + \langle \nabla_{q_s} T(q_s, q_g), u_s^\star \rangle = 0. \tag{14}$$

Note that the optimal policy $u_s^\star$ is a unit vector aligned with the negative gradient $-\nabla_{q_s} T(q_s, q_g)$. Thus, we find that the optimal policy is $u_s^\star = -\nabla_{q_s} T(q_s, q_g)/\|\nabla_{q_s} T(q_s, q_g)\|$. Plugging $u_s^\star$ into Eq. 14 and we arrive at:

$$\frac{1}{S^\star(q_s)} - \|\nabla_{q_s} T(q_s, q_g)\| = 0, \tag{15}$$

which leads to our Eikonal loss $L_E$ at infinitesimal time scale. Additionally, we derive the TD loss as follows:

$$
\begin{aligned}
T(q_s + u_s^\star \Delta t, q_g) &= T(q_s, q_g) + \langle \nabla_{q_s} T(q_s, q_g), u_s^\star \rangle \Delta t + o(\Delta t) \\
&= T(q_s, q_g) - \|\nabla_{q_s} T(q_s, q_g)\| \Delta t + o(\Delta t) \\
&= T(q_s, q_g) - \Delta t / S^\star(q_s) + o(\Delta t),
\end{aligned}
\tag{16}
$$

and our TD loss is derived by dropping the $o(\Delta t)$ and taking $L_2$ norm.

## B  ADDITIONAL RESULTS

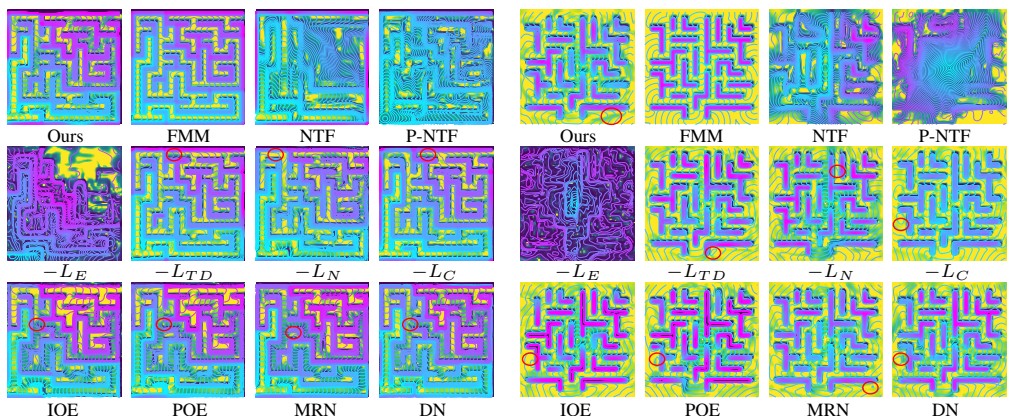

Figure 6: Additonal two mazes results. First row: We compare our method with FMM, NTF, and P-NTF. Middle row: We ablate our method by removing $-L_E$, $-L_{TD}$, $-L_N$, and $-L_C$. Third row: We replace our distance metric $D$ with IQE, PQE, MRN, and DN.

In Fig. 6, we present results for two additional maze environments. For the left maze, our method maintains consistent contour lines without significant artifacts, while NTF and P-NTF fail to recover correct results. All four ablation models exhibit incorrect contour line directions, and other metric formulations also display noticeable artifacts. In the right maze, while our method successfully recovers most of the value function with fewer artifacts compared to the ablations, it struggles to maintain accuracy in the bottom-right corner. Addressing this limitation could be an interesting avenue for future exploration. Finally, Table 2 shows the error of each method to the ground truth solution in the maze scenarios shown in Figs. 3 and 6. The quantitative results confirm that for the left maze, our method achieves lower errors compared to others, as it avoids explicit artifacts present in other approaches. For the right maze, where our method also exhibits some artifacts, the difference is less pronounced but still demonstrates better performance than ablations and comparable results to competing metrics.

In Table 3, we present generalization statistics for both seen and unseen environments. The results indicate that the SR for unseen environments is a little lower compared to seen environments. Additionally, the superiority of our method over existing self-supervised learning approaches is less pronounced than in scalability tasks, primarily due to the increased complexity of the scalability tasks. The drop in SR for self-supervised learning methods further highlights that while generalization is feasible for simpler problems, it becomes significantly more challenging for complex planning problems. As noted earlier, achieving robust generalization to intricate environments, such as those in the Gibson dataset, remains an open research question.

Fig. 7 and 8 provide additional visualizations of paths inferred by our method in Gibson and 12-DOF dual-arm real-world settings.

| Maze | Ours | NTF | P-NTF | $-L_E$ | $-L_{TD}$ | $-L_N$ | $-L_C$ | IQE | PQE | MRN | DN |
|------|------|-----|-------|--------|-----------|--------|--------|-----|-----|-----|-----|
| | | | | | Error of Maze | | | | | | |
| Fig. 3 | 0.08 | 0.58 | 0.58 | 1.13 | 0.21 | 0.13 | 0.12 | 0.32 | 0.46 | 0.19 | 0.29 |
| Fig. 6 (Left) | 0.31 | 1.53 | 1.67 | 7.28 | 0.49 | 0.60 | 0.51 | 0.79 | 0.72 | 0.89 | 0.56 |
| Fig. 6 (Right) | 0.11 | 0.56 | 0.86 | 42.06 | 0.13 | 0.15 | 0.21 | 0.21 | 0.20 | 0.11 | 0.13 |

Table 2: The error of all methods, including ours, on maze environments. The error denotes the mean absolute difference between the travel time of each method and the ground truth FMM, measured at grid points.

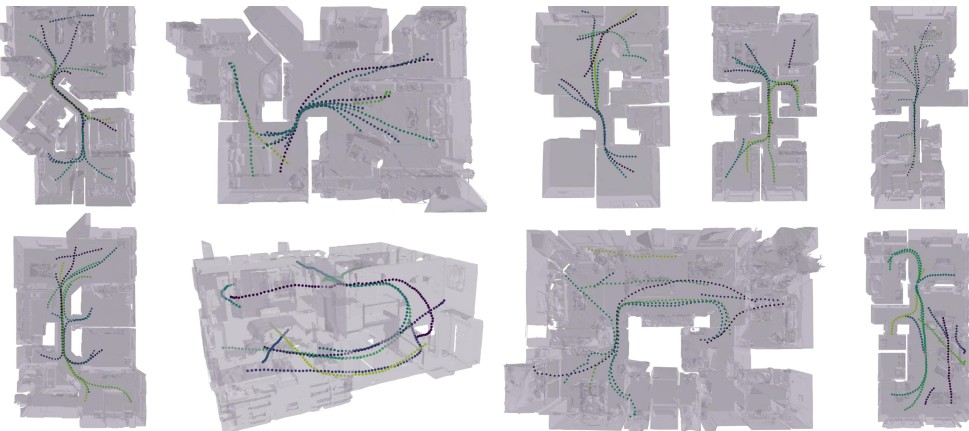

Figure 7: Visualization of Gibson environments, including multiple trajectories planned by our method between various start and goal pairs. It can be observed that our method generates smooth, collision-free trajectories, effectively navigating through the environments while avoiding obstacles.

| Seen | | | | Unseen | | | |
|------|------|------|------|--------|------|------|------|
| (b) Cluttered 3D (C3D) | | | | (b) Cluttered 3D (C3D) | | | |
| Method | Time (s) | Length | SR(%) | Method | Time (s) | Length | SR(%) |
| Ours | $0.026 \pm 0.011$ | $0.71 \pm 0.28$ | 99.6 | Ours | $0.025 \pm 0.011$ | $0.67 \pm 0.30$ | 99.2 |
| NTF | $0.029 \pm 0.010$ | $0.68 \pm 0.26$ | 94.7 | NTF | $0.027 \pm 0.011$ | $0.62 \pm 0.27$ | 92.1 |
| P-NTF | $0.033 \pm 0.032$ | $0.69 \pm 0.27$ | 93.9 | P-NTF | $0.029 \pm 0.013$ | $0.64 \pm 0.27$ | 92.5 |
| MPNet | $0.21 \pm 0.15$ | $0.70 \pm 0.31$ | 96.3 | MPNet | $0.20 \pm 0.17$ | $0.65 \pm 0.31$ | 97.6 |
| FMM | $0.70 \pm 0.09$ | $0.68 \pm 0.25$ | 100 | FMM | $0.73 \pm 0.08$ | $0.63 \pm 0.26$ | 100 |
| RRTC | $0.17 \pm 0.15$ | $0.68 \pm 0.26$ | 100 | RRTC | $0.15 \pm 0.12$ | $0.62 \pm 0.26$ | 100 |
| L-PRM | $0.14 \pm 0.12$ | $0.70 \pm 0.29$ | 100 | L-PRM | $0.13 \pm 0.11$ | $0.65 \pm 0.30$ | 100 |
| (c) 7-DOF Manipulator | | | | (c) 7-DOF Manipulator | | | |
| Method | Time (s) | Length | SR(%) | Method | Time (s) | Length | SR(%) |
| Ours | $0.074 \pm 0.029$ | $1.87 \pm 0.80$ | 88.2 | Ours | $0.075 \pm 0.030$ | $1.95 \pm 0.85$ | 84.0 |
| NTF | $0.067 \pm 0.018$ | $1.63 \pm 0.60$ | 74.6 | NTF | $0.068 \pm 0.020$ | $1.62 \pm 0.69$ | 74.0 |
| P-NTF | $0.063 \pm 0.016$ | $1.62 \pm 0.55$ | 73.3 | P-NTF | $0.066 \pm 0.021$ | $1.73 \pm 0.78$ | 73.3 |
| MPiNet | $0.54 \pm 0.30$ | $2.59 \pm 1.09$ | 92.7 | MPiNet | $0.53 \pm 0.19$ | $2.55 \pm 1.22$ | 90.0 |
| RRTC | $0.42 \pm 0.32$ | $2.02 \pm 1.02$ | 98.0 | RRTC | $0.41 \pm 0.28$ | $2.02 \pm 1.07$ | 97.3 |
| L-PRM | $0.23 \pm 0.46$ | $1.97 \pm 0.93$ | 98.0 | L-PRM | $0.28 \pm 0.57$ | $2.08 \pm 1.20$ | 97.3 |

Table 3: Performance comparison on C3D, and 7-DOF manipulator datasets for seen and unseen environments for our, NTF, and P-NTF methods, while other learning-based methods are not configured for this setting. It can be seen that our method exhibits high SR compared with existing self-supervised learning methods and low planning times.

## C IMPLEMENTATIONS DETAILS

This section summarizes our implementation details, including the training procedure and hyperparameters, as well as details on the data generation and training times of our neural models. In addition to the following details, we aim to publicly release our code on GitHub with the final version of our paper, with which we will also provide our model architectures and facilitate the reproducibility of our method. Furthermore, all experiments and evaluations were conducted on a system with a 3.50GHz × 8 Intel Core i9 processor, 32 GB RAM, and GeForce RTX 3090 GPU.

Start                                Intermediate                                Goal

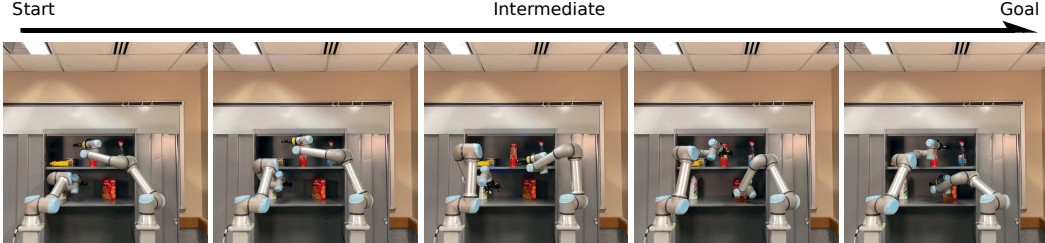

Figure 8: Another path planned by our method to navigate real-world cabinet environment using a 12-DOF dual-arm robot. This trajectory is inferred in 0.1 seconds.

## C.1 TRAINING DETAILS

This section summarizes our training procedure. Our training data comprises randomly sampled robot configurations, their ground truth speed values $S^\star$, and environment point cloud $\mathcal{X}_{obs}$. The ground truth speed values are computed based on the configurations' distance to obstacles using Equation 2. Thus, our method, similar to NTFields, requires only robot configurations and their distance to obstacles, which can be obtained very quickly compared to robot motion trajectories required by supervised learning methods. Next, we randomly from the start, $q_s$, and goal, $q_g$, pairs from the sampled configurations in a given environment. These pair-conditioned latent encodings $f_\theta(q_s, \mathcal{X}_{obs})$ and $f_\theta(q_g, \mathcal{X}_{obs})$ are then obtained followed by the computation of their distance metric $D(f_\theta(q_s, \mathcal{X}_{obs}), f_\theta(q_g, \mathcal{X}_{obs}))$ (Equation 10). This distance represents the travel time $T(q_s, q_g)$ and its gradient with respect to $q_s$ and $q_g$ parameterizes the Eikonal equation (Equation 1) to predict the speed $S(q_s)$ and $S(q_g)$, respectively. The predicted travel-time $T(q_s, q_g)$, its gradients, and corresponding predicted speeds $S(q_s)$ and $S(q_g)$, along with the ground truth speeds $S^\star(q_s)$ and $S^\star(q_g)$, are utilized to compute the loss $L$ (Eq. 8). Finally, we minimize that loss over the sampled data set to train the parameters $\theta$ of our attention-based latent encoders.

## C.2 HYPERPARAMETERS

In 3D environment, we choose $\lambda_E = 10^{-2}, \lambda_{TD} = 10^{-3}, \lambda_N = 10^{-3}, \lambda_C = 0.5$ as the hyperparameters, and we choose TD step $\Delta t = 0.02$. However, for manipulator environments, the free space is much smaller than 3D space, and large TD step and normal direction can lead to the wrong place, so we reduce to $\Delta t = 0.005$, and $\lambda_N = 2 \times 10^{-4}$. We select hyperparameters with the following considerations:

- **Eikonal Loss ($L_E$):** Since $L_E$ involves no approximation, it is expected to be the primary loss term and is assigned the highest weight.

- **Temporal Difference Loss ($L_{TD}$):** $L_{TD}$ utilizes a Taylor expansion around the start and goal points. Its weight is lower than that of $L_E$ to emphasize its complementary role. The choice of $\Delta t$ is crucial; if the value is too large, it may lead to incorrect collision detections in the next state, while a value that is too small diminishes the influence of $L_{TD}$. We determine $\Delta t$ based on the level of clutter in the environment, particularly in narrow passages, to ensure effective value propagation where accurate state transitions are essential. In more cluttered environments, the value of $\Delta t$ needs to be smaller compared to less cluttered ones.

- **Obstacle Alignment Loss ($L_N$):** $L_N$ serves as a guidance loss, encouraging the planning direction to move away from obstacles. However, the true optimal direction should consider both avoiding obstacles and moving toward the goal, which $L_N$ does not fully capture. As such, $L_N$ is assigned a small weight. In narrow-passage environments, where pure obstacle avoidance might conflict with the correct planning direction, a lower weight ensures that $L_N$ does not dominate the loss function.

- **Causality Loss ($L_C$):** For $L_C$, if $\lambda_C$ is too small, its impact on the overall loss is minimal. Conversely, if $\lambda_C$ is too large, it can cause the value function $T(q_s, q_g)$ to grow excessively. We select $\lambda_C$ via cross-validation to ensure stable training.

## C.3 DATA GENERATION AND TRAINING TIMES

Table 4 provides our data generation and training time. It can be seen that the data generation times for our self-supervised method range from a few seconds to minutes. It should be noted that the data generation times for supervised learning methods such as MPNet can take several hours compared to our few minutes.

| Env | Data Generation Time | Training Time |
|---|---|---|
| Maze | 2.9s | 500 epochs 94s |
| Gibson | 3s | 5000 epochs 9min |
| Dual UR5 | 200s | 9500 epochs 32min |
| Cluttered 3D | 100×2s | 2000 epochs 40min |
| Franka | 150×20s | 2000 epochs 46min |

Table 4: Data generation and training time

