# OpenReview forum: "Physics-informed Temporal Difference Metric Learning for Robot Motion Planning"
_ICLR.cc/2025/Conference — ICLR 2025 Poster_

### Official Review · Reviewer_qf75 · 2024-11-01

**Soundness:** 3
**Presentation:** 3
**Contribution:** 3
**Rating:** 8
**Confidence:** 3

**Summary:**

The paper presents a self-supervised learning approach for robotic motion planning that is based on learning to approximate the Eikonal equation. The work addresses a key challenge in robotics, that of navigating complex environments without requiring costly, expert-generated training data. The authors propose a method that used the Eikonal equation to learn a cost-to-go function that represents both an optimal value function and a geodesic distance in the planning/configuration space.

**Strengths:**

The paper is well written and clearly presents the approach and evaluation on a set of relevant and reasonable datasets.

The method outperforms existing methods in terms of computational speed, with good performance in terms of success rate, and path length.

The main strength of the approach is the computational efficiency and that it generalizes to unseen environments, which according to the authors was not possible previously with prior self-supervised methods.

The approach is easy to setup and fast to train (few seconds to few minutes), though there are nuances about hyper-parameters as the authors point out.

The ablation part of the paper gives a clear picture of what and how contributes to the methods performance.

**Weaknesses:**

The main weakness of teh paper is the lack of quantitative analysis of the generalisation capability of the approach. The authors only present visualisations of successful paths in unseen environments. A more comprehensive analysis would be more appropriate, for example,  evaluating the performance across a wide range of unseen environments with varying complexities and against the sota approaches. This would provide a more objective measure of robustness/SR and identify specific types of environments where the approach might struggle.

**Questions:**

For the ablation part, is it possible that using only one space configuration might be biasing your evaluations? Have you tried testing on different maze configurations or other types of spaces?

The claim of achieving hight SR in the comparisons in Table 1 could be made less strong or reworded. The SR of the method is consistently on par or lower from the top SR in all examples.

For table 1, clarify what you measure for time in the different cases.

Expand on and give examples on the failure cases you mention when generalizing to unseen environments.

"Superior speed" in s5.3, maybe reword to computational performance or similar.

---

> ### Author Response · Authors · 2024-11-17
> **For Reviewer qf75**
>
> We sincerely thank the reviewer for the comprehensive evaluation and response to the major concerns below.
>
> **1.** *lack of quantitative analysis of the generalization capability of the approach. A more comprehensive analysis would be more appropriate, for example, evaluating the performance across a wide range of unseen environments with varying complexities and against the Sota approaches.*
>
> **Authors:** We have added our generalization setup to NTFields and P-NTFields to show the ability of our method compared with previous physics-informed methods. In addition, we have now also separately reported the results of seen and unseen experiments in Appendix B.
>
> **2.** *For the ablation part, is it possible that using only one space configuration might be biasing your evaluations? Have you tried testing on different maze configurations or other types of spaces?*
>
> **Authors:**  We have included more complex mazes in Appendix B. The results are consistent with what we reported in a single maze environment. Additionally, we would like to highlight that our experiments in Gibson, cluttered 3D environments, and 7 DOF manipulations can be considered ablation studies when compared to NTFields and P-NTFields. The previous methods do not utilize TD Loss, obstacle alignment loss, or causality loss, and they have been shown to perform poorly compared to our methods. Furthermore, these previous methods do not function at all in the 12 DOF manipulation environment, which further validates the significance of our proposed methods and their novel objective functions.
>
> **3.** *The claim of achieving high SR in the comparisons in Table 1 could be made less strong or reworded. The SR of the method is consistently on par or lower from the top SR in all examples. "Superior speed" in s5.3, maybe reword to computational performance or similar.*
>
> **Authors:** We thank reviewers for these suggestions and we have made the suggested changes as highlighted in the paper.
>
> **4.** *For table 1, clarify what you measure for time in the different cases.*
>
> **Authors:** We report computational time representing wall clock time taken by a CPU-based execution of all presented planners. We have clarified it in Section 5.
>
> **5.** *Expand on and give examples of the failure cases you mention when generalizing to unseen environments.*
>
> **Authors:**  Currently, our method generalizes to relatively simple, unseen scenarios, such as 3D box environments and 7 DOF manipulations in a table-top setting. However, generalizing to unseen complex environments e.g. the Gibson dataset remains a challenging and unresolved issue. We believe this difficulty arises from the varying intricate obstacle geometries present in different Gibson scenarios, which are hard to represent within our current training set and model architecture. Therefore, in our future work, we aim to enhance the generalization ability of our method for complex scenarios with intricate obstacle geometries.

---

> ### Author Response · Authors · 2024-11-22
> **Looking forward to discussion and reassessment of our work**
>
> Dear Reviewer,
> thank you for your valuable feedback. We have revised the manuscript and provided detailed responses to address your concerns.
> We hope our revisions meet your expectations and kindly request your reassessment.
> Thank you for your time and consideration.

---

### Official Review · Reviewer_1DdR · 2024-11-02

**Soundness:** 3
**Presentation:** 3
**Contribution:** 3
**Rating:** 8
**Confidence:** 3

**Summary:**

This paper introduces a self-supervised learning approach for robot motion planning. It builds upon prior work on physics-informed neural motion planning, namely NTFields, in order to tackle complex cluttered environments. This is done through the use of temporal difference (TD) learning to solve the Eikonal equation as an optimal value function, which enforces Bellman’s principle of optimality and enhances convergence. The authors also propose a novel neural network architecture for metric learning that preserves the properties of geodesic distance, and ensure that the triangle inequality, symmetry, and non-negativity are respected. This model is used to compute the cost-to-go in sampling based Model Predictive Control (MPC) to perform motion planning. Experiments are conducted using robots with different numbers of DOFs on various types of environments, and results support most of the authors claims.

**Strengths:**

- The paper tackles an important problem in robotics and does a good job of highlighting the challenges of performing motion planning in complex cluttered environments for high DOFs robots.
- The authors provide a comprehensive literature review and the approach proposed is well positioned compared to prior work in traditional and neural motion planning.
- The proposed method is well thought-out and well detailed in the paper. The design choices are sufficiently justified.

**Weaknesses:**

- The experimental methodology suffers from some weaknesses. First, the ablation study is conducted on a single 2D maze environment. While results highlight the importance of each loss and the chosen metric for this specific environment, they do not necessarily show that this importance/performance is maintained across different environments. Second, in the generalization to novel environments, both seen and unseen environments are used during testing, which in my opinion defeats the purpose of testing the generalization to unseen environments.
- The article is too technical and a bit complex to follow for a reader who is not completely familiar with this class of approaches. Moreover, the figures are hard to decrypt, captions should provide more details on how to read the figures with legends depicting what each element represents.
- The method uses a set of user-defined hyperparameters, their values are stated in Apprendix C, however, the paper does not provide any comment on the sensitivity of the approach to these hyperparameters and how to mitigate it, especially the $\Delta T$ parameter.

**Questions:**

- Is there a reason why the ablation study was not done in a quantitative manner showcasing that the obtained results are consistent across different environments ?
- Why are both seen and unseen environments used during testing for the evaluation of the generalization to novel environments ?
- In Fig.1, what do the green points represent ? What about the black line ?
- In Fig.2, what does the color and the contour lines represent ?
- Why is the success rate lower for the 7DOF manipulator at 87% compared to the other experiments, especially since for the 12DOF dual-arm, the method achieves a 91% success rate ?
- Have you conducted a sensitivity analysis to measure the importance of the choice of hyperparameters, and how to set them for each case ?

---

> ### Author Response · Authors · 2024-11-17
> **For Reviewer 1DdR (Part 1)**
>
> We sincerely thank the reviewer for the comprehensive evaluation and response to the major concerns below.
>
>
> **1.** *The ablation study is conducted on a single 2D maze environment*
>
> **Authors:** We have included more complex mazes in Appendix B. The results are consistent with what we reported in a single maze environment. Additionally, we would like to highlight that our experiments in Gibson, cluttered 3D environments, and 7 DOF manipulations can be considered ablation studies when compared to NTFields and P-NTFields. The previous methods do not utilize TD Loss, obstacle alignment loss, or causality loss, and they have been shown to perform poorly compared to our methods. Furthermore, these previous methods do not function at all in the 12 DOF manipulation environment, which further validates the significance of our proposed methods and their novel objective functions.
>
> **2.** *In the generalization to novel environments, both seen and unseen environments are used during testing*
>
> **Authors:** In Appendix B, Table 2, we have included the generalization statistics for both seen and unseen environments separately. The results indicate that the success rate (SR) for unseen environments is only slightly lower than that for seen environments, while the overall performance remains similar. This demonstrates our method's ability to generalize to unseen tasks. Additionally, we have extended the prior methods, NTFFields, and P-NTFields, with our attention-based architecture to enable their generalization across multiple environments, as their original architecture was not capable of scaling to accommodate multiple environments.
>
> **3.** *The article is too technical and a bit complex to follow for a reader who is not completely familiar with this class of approaches. Moreover, the figures are hard to decrypt, captions should provide more details on how to read the figures with legends depicting what each element represents.*
>
> **Authors:** We have revised our explanations for Figures 1 and 3 (previous 2), as these two figures are critical for understanding our method. Please let us know if this helps the reviewers understand our approach.
>
> **4.** *Have you conducted a sensitivity analysis to measure the importance of the choice of hyperparameters, and how to set them for each case?*
>
> **Authors:**  We have now included guidelines for tuning hyperparameters in our revised Appendix C. Following these guidelines makes the hyperparameter selection process quite straightforward.
>
>
> **5.** *Is there a reason why the ablation study was not done in a quantitative manner showcasing that the obtained results are consistent across different environments?*
>
> **Authors:** We include Fig. 3 (previous Fig. 2) as it is crucial to understand the **contour lines** in that figure. These lines represent points with the same travel time function value, $T(q_s,q_g)$, and their normal direction indicates the optimal motion path toward the goal configuration. In a globally optimal value function, contour lines should not exhibit local minima. In Fig. 3 (previous Fig. 2), we observe that the contours produced by other methods contain spurious local minima or the normal direction penetrates the solid boundary, whereas our method eliminates these artifacts.
>
> It’s important to note that the loss function measures the correctness of contour lines across the entire configuration space, while spurious local minima are localized phenomena. As a result, the improvement in loss may not appear substantial. However, this does not diminish the importance of the difference. Even a single spurious local minimum can trap the robot in its vicinity, preventing it from reaching the true goal configuration. We have revised our description in Section 5.1 to highlight the importance of correct contour lines while having low errors.
>
> **6.** *Why are both seen and unseen environments used during testing for the evaluation of the generalization to novel environments?*
>
> **Authors:** We highlight two types of unseen factors in our learning task: **unseen start-goal configurations** and **unseen environments**. In all tests, whether in seen or unseen environments, we consistently use unseen start-goal configurations. Testing in unseen environments ensures that our learned value function can handle any motion planning query within a newly encountered environment.

---

> ### Author Response · Authors · 2024-11-17
> **For Reviewer 1DdR (Part 2)**
>
> **7.** *In Fig.1, what do the green points represent? What about the black line?*
>
> **Authors:** We revised Fig. 1 and its caption to better illustrate the example problem. In summary, to understand this figure, note that the Eikonal equation we want to solve is: $|T’(q)|=1, T(0)=0$. Our Eikonal loss, $L_E=(|T’(q)|-1)^2$, is evaluated at discrete sample points (green). The top figure shows the correct solution $T(q)=|q|$, and the bottom figure has spurious local minima. If we rely on $L_E$, on all sampled green points, we have $L_E=0$ for both plots. However, suppose we use $L_{TD}$. In that case, we can enforce for some point $q_1>0$, we have $T(q_1)-T(q_1-d)=d$ for a local perturbation $d>0$. This leads to a higher loss for the bottom solution than the top solution, which aligns with our desired behavior. We have updated the figure caption to reflect these points more clearly.
>
>
> **8.** *In Fig.2, what does the color and the contour lines represent?*
>
> **Authors:** The color shows the speed field, which is also the Truncated SDF of the obstacle, and we are training based on this speed field. The contour lines consist of points with the same travel-time function value $T(q_s,q_g)$. Following the negative gradient direction of the value function (normal direction of the counter line), we can find a globally optimal path leading to the goal configuration. We have briefly included that description in Section 5.1
>
> **9.** *Why is the success rate lower for the 7DOF manipulator at 87% compared to the other experiments, especially since for the 12DOF dual-arm, the method achieves a 91% success rate?*
>
> **Authors:** (Please refer to Answer 6) We appreciate the reviewer’s observation. The difference in success rates arises from the nature of the experiments. The 12-DOF dual-arm experiment focuses on scalability within a seen environment and unseen start and goal pairs. In contrast, the 7-DOF manipulator experiment evaluates the method’s ability to generalize to unseen environments, which poses a significantly greater challenge. This highlights the inherent trade-off between scalability in fixed settings and generalization across diverse scenarios.

---

> ### Author Response · Authors · 2024-11-22
> **Looking forward to discussion and reassessment of our work**
>
> Dear Reviewer,
> thank you for your valuable feedback. We have revised the manuscript and provided detailed responses to address your concerns.
> We hope our revisions meet your expectations and kindly request your reassessment.
> Thank you for your time and consideration.

---

> > ### Comment · Reviewer_1DdR · 2024-11-25
> > **Response to authors**
> >
> > Thank you for your detailed and thoughtful rebuttal. The revised manuscript is now much clearer and easier to follow. The separation of results between seen and unseen environments is more sound and effectively demonstrates the proposed method's ability to generalize to new environments.
> >
> > However, I still have concerns about the choice to perform the ablation study in a qualitative manner. While I understand the value of Figure 3 and acknowledge that other experiments could serve as ablation studies to some extent, they do not quantitatively highlight the contribution of each loss term. Including a table summarizing the errors obtained when each loss term is ablated across different environments would significantly enhance the paper's clarity and impact.
> >
> > That said, I recognize that time constraints may limit the possibility of conducting additional experiments. If this is the case, I would suggest simply reporting the numerical errors for the 2D mazes included in Appendix B to allow comparison with the results reported in section 5.1.

---

> > > ### Author Response · Authors · 2024-11-25
> > > **Thank you very much for engaging in the discussion**
> > >
> > > Thank you for your thoughtful feedback and for engaging in this constructive discussion. Your insights have been invaluable in improving the clarity and rigor of our work.
> > >
> > > To address your concern regarding the quantitative ablation study, we have summarized the numerical errors of all methods in the 2D maze environments from Appendix B. The table is organized in the same order as the figure: the first row shows results for our method, FMM, NTF, and P-NTF; the second row presents results for ablations (without Eikonal loss, without TD loss, without normal alignment loss, and without causality loss); and the third row includes results for other metrics (IQE, PQE, MRN, and DN).
> > >
> > > Left Maze Results:
> > > |     |  |    |    |
> > > | ---    | ---   | ---     |---     |
> > > | 0.31  | 0   | 1.53  | 1.67|
> > > | 7.28| 0.49| 0.60|0.51|
> > > | 0.79| 0.72| 0.89| 0.56|
> > >
> > > Right Maze Results:
> > > |     |  |    |    |
> > > | ---    | ---   | ---     |---     |
> > > | 0.11  | 0   | 0.56 | 0.86|
> > > | 42.06| 0.13| 0.15|0.21|
> > > | 0.21| 0.20|  0.11| 0.13|
> > >
> > > The quantitative results confirm that for the left maze, our method achieves lower errors compared to others, as it avoids explicit artifacts present in other approaches. For the right maze, where our method also exhibits some artifacts, the difference is less pronounced but still demonstrates better performance than ablations and comparable results to competing metrics.
> > >
> > > We hope these results address your concerns and further highlight the contributions of each component in our approach. Thank you again for your engagement and thoughtful discussion throughout this process.

---

> > > > ### Comment · Reviewer_1DdR · 2024-11-25
> > > > **Thank you for your response**
> > > >
> > > > I appreciate the authors' time and efforts throughout the discussion phase. Most of my concerns have been addressed. I am considering raising my score, under the condition that the tables/ablation results above are included in the paper.

---

> > > > > ### Author Response · Authors · 2024-11-25
> > > > > **Response to reviewer**
> > > > >
> > > > > Dear Reviewer, Thank you very much for your valuable time and feedback. We have included the table in the paper; please refer to Table 2 in Appendix B. We also appreciate your consideration in raising the score. If you have any additional questions, please let us know.

---

> > > > > > ### Comment · Reviewer_1DdR · 2024-11-25
> > > > > >
> > > > > > The revised manuscript is clear, sound and addresses all my concerns. Therefore I have updated my score.

---

### Official Review · Reviewer_7oHd · 2024-11-04

**Soundness:** 3
**Presentation:** 3
**Contribution:** 3
**Rating:** 5
**Confidence:** 4

**Summary:**

This paper improves a prior physics-based neural motion planning method by proposing a new metric learning approach to better approximate the solution of the Eikonal equation. This is done by enforcing Bellman’s principle of optimality based on the observation that the solution to the Eikonal equation can be viewed as an optimal value function. Further, a new network architecture is proposed for generalizable metric learning. This approach is claimed to perform better than existing motion planning and learning-based planning methods.

**Strengths:**

- The method is significantly faster than sampling based motion planning.

- Compared to other learning-based planning methods that require an expensive data collection phase, this method is self-supervised and hence removes that requirement.

**Weaknesses:**

- The proposed method has worse success rates than sampling-based planners. In particular, in the harder 7-DOF manipulator experiment, the proposed method is significantly worse (87%) than learning-free RRT-Connect (98%).

- I am not convinced by the claim in the abstract that the method significantly outperforms existing methods, particularly in the more complex 7-dof domain. While it is much faster and has decent success rates, the results are more mixed.

- The overall loss function has many hyperparameters.

**Questions:**

- Why is solving Eikonal equation more scalable than motion planning? What approximations or relaxations does it make? Does it satisfy the robot’s kinodynamic constraints?

- The MPC scheme being proposed for planning with the learned value function cannot back-track. Wouldn’t it be better to use a search algorithm, e.g. A\* with the value function as a heuristic to guarantee completeness. The proposed MPC-based planning may fail to solve hard planning problems which is somewhat supported by the low SR in 7-dof planning.

- In the 7-dof experiment, what is RRTC’s planning time on the problems solved successfully by the proposed method. It is possible that RRTC’s planning time is higher because it is solving harder problems that your methods fails at. For a fair comparison, it would be useful to know the average planning time for problems successfully solved by all methods.

- How did you tune the loss hyperparameters?

---

> ### Author Response · Authors · 2024-11-17
> **For Reviewer 7oHd**
>
> We sincerely thank the reviewer for the comprehensive evaluation and response to the major concerns below.
>
> **1.** *Lower success rate than Sampling-based Motion Planners (SMP).*
>
> **Authors:** We acknowledge that, given sufficient computational time (up to 30 seconds in our evaluation), NTFields-type methods currently cannot achieve the same success rate as Sampling-based Motion Planners (SMPs) like RRT-Connect. This limitation affects all existing learning-based methods, and ongoing research in learning-based motion planning aims to address these challenges, including our work. In contrast, many SMPs are probabilistically complete and are expected to achieve a high success rate, as they will eventually converge to find a feasible or optimal solution if one exists. However, our method, like other learning-based approaches, offers significantly faster inference times than SMPs, making it far more suitable for time-sensitive, real-time control applications.
>
> Regarding the incorporation of our approach with Model Predictive Control (MPC): The proposed method provides a value function by solving the Eikonal equation. This value function can be integrated into any downstream planner, such as RRT, $A^\star$, MPC, or trajectory optimization. We use MPC as a demonstration of this flexibility. Additionally, these downstream planners can accommodate extra costs, such as collision constraints that can be layered on top of the original value function to improve robustness.
>
> **2.** *Hyperparameters need tuning*
>
> **Authors:** Our method needs some hyperparameters, but their tuning guideline has been provided in our revised Appendix C. Under these guidelines, we discuss that tuning hyperparameters in our experiments is a trivial task
>
> **3.** *What is the benefit of solving the Eikonal equation for motion planning?*
>
> **Authors:** The primary advantage of using a neural Eikonal equation solver is its ability to quickly infer the globally optimal direction of motion to reach a goal configuration. In other words, it enables real-time responses to planning queries. In contrast, Sampling-based Motion Planners (SMP) are guaranteed to be probabilistically complete, that is, they will always find a feasible or optimal solution, but they often require significant computational effort for each query.
>
> The Eikonal equation solution indicates the shortest travel time and defines the geometrically shortest path between two locations. The key assumption underlying the neural Eikonal solver is that the optimal value function can be effectively approximated by a neural network. While this assumption generally holds, it can be violated, leading to spurious local minima, as shown in Fig. 3 (previous Fig. 2). Nonetheless, our method demonstrates substantial improvements over previous approaches to solving the Eikonal equation.
>
> As the reviewer noted, a limitation of our method is its inability to handle kinodynamic constraints directly. We have acknowledged this in the limitations section. However, our value function can be effectively integrated as a cost-to-go function within kinodynamic MPC frameworks. Notably, prior work [1] has addressed this challenge by leveraging a precomputed RRT tree as a terminal cost, and similar strategies could be applied to extend our method for handling kinodynamic constraints.
>
> [1] The Value of Planning for Infinite-Horizon Model Predictive Control
>
> **4.** *MPC is not good and $A^\star$ search is better.*
>
> **Authors:** By integrating the learned travel-time function with online MPC, we effectively generalize the concept of travel time to a more versatile value or cost-to-go function. This approach provides a flexible framework for addressing planning problems. For example, we could leverage completeness-guaranteed search algorithms, such as $A^\star$ or T-RRT [1], guided by our learned value function. However, combining the value function with different search frameworks is beyond the scope of this work. Our primary focus is on learning a good value function.
>
> Each search framework offers distinct advantages. Online MPC can quickly escape local minima in the value function but performs only local searches and lacks completeness. In contrast, methods like T-RRT and $A^\star$ guarantee completeness but require more computational time. If desired, we could conduct experiments demonstrating the performance of our approach when combined with these alternative search frameworks.
>
> [1] Sampling-Based Path Planning on Configuration-Space Costmaps
>
> **5.** *What is the running time for RRTC for successful cases solved by our method*
>
> **Authors:** In the manipulation environment, for only successful cases the RRTC statistics are: Time $0.37 \pm 0.24$, Length $1.83 \pm 0.92$ and L-PRM statistics are: Time $0.15 \pm 0.42$, Length $1.84 \pm 0.94$. Note that these times are still significantly higher than our approaches in general.

---

> > ### Comment · Reviewer_7oHd · 2024-11-22
> > **Response to authors.**
> >
> > Thanks a lot for responding to my questions. However, I still think that the paper does not accurately and clearly describe what solving the Eikonal equation provides. It is my understanding that it provides a heuristic, i.e., an approximate cost-to-go and that the approach is incomplete. In this light, the response, "The primary advantage of using a neural Eikonal equation solver is its ability to quickly infer the globally optimal direction of motion to reach a goal configuration" seems inaccurate since no guarantee of global optimality is provided by the approach.

---

> > > ### Author Response · Authors · 2024-11-22
> > > **Thank you very much for your response**
> > >
> > > Thank you very much for your response. We would like to clarify two important points: the nature of the Eikonal equation’s solution and the guarantees of a solver for the Eikonal equation.
> > >
> > > First, as detailed in Appendix A and supported by prior works [1,2], the solution to the Eikonal equation corresponds to the shortest path and represents a globally optimal solution in the mathematical sense. This solution inherently encodes the globally optimal cost-to-go function in free space with respect to the defined speed field.
> > >
> > > Second, while the theoretical solution to the Eikonal equation is globally optimal, a neural network-based solver like ours does not guarantee finding the exact optimal solution due to approximations and potential limitations of the learned model. That said, our method demonstrates significant improvements over previous neural Eikonal equation solvers in approximating this solution, achieving better results in both scalability and generalization.
> > >
> > > Additionally, it is worth emphasizing that the globally shortest path encoded by the Eikonal equation’s solution is not merely a heuristic—it is the actual global cost-to-go function. In contrast, simpler heuristics, such as Euclidean distance used in A* search, may lead to incorrect search directions in complex scenarios. By leveraging a global guidance mechanism, our approach provides better planning capabilities compared to traditional heuristic-based methods.
> > >
> > > [1] A fast marching level set method for monotonically advancing fronts
> > >
> > > [2] NTFields: Neural Time Fields for Physics-Informed Robot Motion Planning

---

> > > > ### Author Response · Authors · 2024-11-25
> > > > **Request for discussion or reassessment of our work**
> > > >
> > > > Dear Reviewer, Thank you once again for your valuable feedback. This is a gentle reminder that the final date for discussions between reviewers and authors is November 27. If you have any further questions, please let us know. We have revised the manuscript and provided detailed responses to address your concerns. We hope our revisions meet your expectations and kindly request your reassessment. Thank you for your time and consideration.

---

> > > > > ### Comment · Reviewer_7oHd · 2024-12-01
> > > > > **Final recommendation.**
> > > > >
> > > > > Dear authors,
> > > > > Thank you for your active participation in the rebuttal process and addressing some of my concerns. Unfortunately, I won't be increasing my score. I believe it is an interesting direction and paper. However, the lack of guarantees and large number of hyperparameters make the approach less appealing. I believe the approach would be much stronger when combined with a complete motion planner.

---

> > > > > > ### Author Response · Authors · 2024-12-01
> > > > > > **Response to reviewer**
> > > > > >
> > > > > > We thank the reviewer for the feedback. Completeness in motion planning is a well-studied topic, and as shown in methods like A* and T-RRT [1], it is typically ensured through sampling and search techniques. Our contribution is not about reintroducing completeness guarantees but rather about learning a high-quality cost function that significantly improves efficiency and achieves a high success rate without relying on a complete planner. Adding a completeness guarantee using an existing method like T-RRT would not add to or detract from our contribution—it would simply build on the foundation we have already established. The results we present demonstrate that even without such guarantees, our approach performs effectively and efficiently. We think the reviewer has misinterpreted key aspects.
> > > > > >
> > > > > > [1] Sampling-Based Path Planning on Configuration-Space Costmaps

---

> ### Author Response · Authors · 2024-11-22
> **Looking forward to discussion and reassessment of our work**
>
> Dear Reviewer,
> thank you for your valuable feedback. We have revised the manuscript and provided detailed responses to address your concerns.
> We hope our revisions meet your expectations and kindly request your reassessment.
> Thank you for your time and consideration.

---

### Official Review · Reviewer_3RM5 · 2024-11-08

**Soundness:** 3
**Presentation:** 2
**Contribution:** 3
**Rating:** 6
**Confidence:** 3

**Summary:**

This paper proposes a new method to allow robots to use self-supervised learning to learn motion planners and use MPC for online path generation.

The proposed method is built on NTFields (Ni & Qureshi, 2023a), and proposes two improvements:
1. Novel loss
    1. View the self-supervised learning problem in NTFields as a solution to an optimal control problem, propose a discrete-time loss L_TD based on temporal difference learning in RL, and combine L_TD with NTFields's continuous-time loss L_E to capture both a coarse-grained view of value propagation and a fine-grained optimal physics structure.
    2. Another loss term L_N to avoid collision.
    3. Another loss term L_C to ensure causality (i.e., time only moves forward) based on (Wang et al., 2024b).
2. Metric learning: during NTFields's training, the structure of the learned travel-time function might violate triangle inequality, becoming no longer a geodesic distance on a Riemannian manifold. This paper proposes a neural network structure that preserves the key properties of such a metric.
3. Allow generalization to the new environments by conditioning the learning on the environment via the attention mechanism.
4. Use online MPC for path generation.

**Strengths:**

- The problem is well-motivated.
- Very clear discussion about related works, including traditional motion planning methods, learning from experts, and self-supervised learning.
- (Strength and also some weakness) I think the first improvement, using both continuous loss L_E from previous work and a novel TD loss is interesting. There is some discussion about the benefits of combining fine-grained and coarse-grained structures. The authors also used an example (Fig.1) to show that sometimes, only ensuring a low L_E does not imply good behavior. However, I don't quite get why L_E is 0 in both cases and why L_TD is high in the 2nd case. It would be great if the authors could provide more explanation about Fig.1. I think this could make the contribution more significant.
- The result is very comprehensive, with simulation and real-world benchmarks, and compared against many baseline methods.

**Weaknesses:**

- I think the key weakness is that even though the various novel techniques improve NTFields, the empirical result does not seem to convey that impact. For example, in the ablation study (Fig.2 row 2), the errors discussed in line 403 convey that L_E, which was proposed in NTFields, is the most important component, and the other loss terms only provide small improvements (e.g., from 0.21 to 0.08 error). This makes me worry a bit about the effectiveness of the proposed techniques. In the real world, the proposed techniques increase the system's complexity, so it would be great if they provide a very strong performance boost to motivate people to adopt them. It would be great if the authors could discuss the importance of the proposed techniques, perhaps explaining how to interpret these results or using some other qualitative results to demonstrate.
    - Related to this, it seems that one major improvement of the proposed method compared to previous work in Tab.1 is faster plan generation, e.g., from 0.5 sec to 0.07 sec. I think it would be great if the authors could discuss the implications of why this could matter in the real world. Perhaps one way to show that this is significant is to consider scenarios where the robot has to keep replanning to adapt to dynamic obstacles.

- I think this paper did a good job introducing the technical key points about the previous work NTFields (Ni & Qureshi, 2023a) in Sec.3.2. However, I think it lacks some high-level description of how NTFields works as a motion planner. As a result, as a person outside this field of self-supervised motion planning, I found Sec.3.2 a bit hard to follow. For example, here are some points where I got confused:
    - It seems that S*(q) is a function that is manually defined that outputs the desired robot velocity when the robot is around obstacles.
    - Eq.3 seems to propose training the function S such that it matches S* at starting and goal configurations. Then, a perhaps naive question is why to learn S rather than just using S* as the motion planner?
    - How is the learned S used to generate motion plans during online inference time? Line 352 mentions that some previous methods do gradient descent to minimize T(qs,qg). So, I guess it is looking for some path that minimizes the time from qs to qg? It would be great to put this information earlier in Sec.3.2.
- Similar to the previous point, I think more intuition can help Sec.4.2.
    - I think the motivation for metric learning in Sec.4.2 is not so clear to me. I notice that the authors briefly mentioned the potential benefit of metric learning in line 114 by saying, "scale and generalize effectively in complex, cluttered, or unseen environments." But it is not so intuitive. Would there be some intuitive example, such as Fig.2, that can illustrate why metric learning is important?
    - Eq.10 seems to be the key contribution. However, the discussion around it from lines 304 to 313 is very abstract. It would be great if the authors could provide some high-level intuition about why L_\inf distance is preferred to Euclidean distance.
- The contribution of Sec.4.3 - online MPC is interesting. It seems that the key benefit of this is faster online inference compared to (Ni et al., 2021; Ni & Qureshi, 2023b). However, the empirical results in Tab.1(a) show that the proposed method's computation time is 0.056 while NTF (Ni et al., 2021) is 0.074, with a small difference. Could the authors discuss the potential implications of why this difference could matter in the real world and how it will strengthen the contribution?
- Fig.2 is a key result. However, for people outside the field, it is unclear how to interpret it. It would be great to add some guidance.


## Minor
- In Sec.3.1, it might be helpful to mention that Q_free and Q_obs are assumed to be known. I think this might be an implicit assumption for motion planning. But in reality, these things have to be obtained from sensors, which is usually nontrivial. So, it might be helpful to mention the assumption for readers outside motion planning.
- Line 331 suddenly mentions "the attention mechanism," which seems interesting and important. It would be great to elaborate on why using "attention" so that I, as someone who is not familiar with attention, can also appreciate the contribution.

**Questions:**

- Sec.2 discusses the relationship between RL and self-supervised learning using physics-informed neural networks. I wonder about the difference between them. Can I think of a self-supervised learning type of approach as doing offline RL but with specially designed network architecture? The reason why I ask this question is that I think Sec.3.1 and 3.2 could have some introduction about how self-supervised learning motion planners are trained. I think this could be very helpful for people who are not familiar with NTFields (Ni & Qureshi, 2023a).
- Sec.4.1.2 designs another loss for avoiding collisions. What is the relation between that and LE, where I think S* is also avoiding collisions based on the discussion around Eq.2?

---

> ### Author Response · Authors · 2024-11-17
> **For Reviewer 3RM5 (Part 1)**
>
> We sincerely thank the reviewer for the comprehensive evaluation and response to the major concerns below.
>
> **1.** *Provide more explanation about Fig.1.*
>
> **Authors:** To understand this figure, note that the Eikonal equation we want to solve is: $|T’(q)|=1, T(0)=0$. Our Eikonal loss, $L_E=(|T’(q)|-1)^2$, is evaluated at discrete sample points (green). The top figure shows the correct solution, and the bottom figure has spurious local minima.
>
> If we rely on $L_E$, on all sampled green points, we have $L_E=0$ for both plots. However, suppose we use $L_{TD}$. In that case, we can enforce for some point $q_1>0$, we have $T(q_1)-T(q_1-d)=d$ for a local perturbation $d>0$. This leads to a higher loss for the bottom solution than the top solution, which aligns with our desired behavior. We have updated the figure caption to reflect these points more clearly.
>
> **2.** *The empirical result does not significantly improve over NTFields.*
>
> **2a.** *In Fig.2 (the maze), $L_E$ is the most important component, other loss terms only provide small improvements.*
>
> **Authors:** To appreciate the significance of our improvement, it is crucial to understand the **contour lines** in Fig. 3 (previous Fig. 2). These lines represent points with the same travel time function value, $T(q_s,q_g)$, and their normal direction indicates the optimal motion path toward the goal configuration. In a globally optimal value function, contour lines should not exhibit local minima. In Fig. 3 (previous Fig. 2), we observe that the contours produced by other methods contain spurious local minima or the normal direction penetrates the solid boundary, whereas our method eliminates these artifacts.
>
> It’s important to note that the loss function measures the correctness of contour lines across the entire configuration space, while spurious local minima are localized phenomena. As a result, the improvement in loss may not appear substantial. However, this does not diminish the importance of the difference. Even a single spurious local minimum can trap the robot in its vicinity, preventing it from reaching the true goal configuration. We have revised our description in Section 5.1 to highlight the importance of correct contour lines while having low errors.
>
>
> **2b.** *Why is planning fast important?*
>
> **Authors:** In the real world, fast planning is crucial for realizing a wide range of real-time robotic applications in fields such as smart cars, robotic surgery, UAVs (unmanned aerial vehicles), and robot-assisted manufacturing. As the reviewer pointed out, our training assumes arbitrary start-goal configurations for the robot, enabling zero-shot responses to any planning query, which enhances its applicability across all the domains mentioned above. Additionally, the Eikonal equation solution provides a global value function. It can be incorporated into any off-the-shelf planning strategies such as MPC and trajectory optimization. These downstream planners can deal with scenarios including dynamic obstacles guided by our learned value function. Nevertheless, such analysis is beyond the scope of this paper.
>
> **3.** *Clarification about paper writing*
>
> **3a.** *Since $S^\star$ is manually defined, why learn $S$ rather than use $S^\star$ for planning? Eq.3 seems to propose training the function S such that it matches $S^\star$ at starting and goal configurations. Then, a perhaps naive question is why learn $S$ rather than just using $S^\star$ as the motion planner? How is the learned $S$ used to generate motion plans during online inference time?*
>
> **Authors:** Please note that  $S$ or $S^\star$ cannot be directly used for motion planning as they simply provide a scalar value of a given configuration based on its distance to the obstacle. The Eikonal equation provides a way to relate this speed function to the travel time function $T$. Our objective is to learn $T$ in a self-supersized manner by utilizing $S^\star$. Our network prediction is $T$ and we compute $S=1/\|\nabla T\|$ and the $L_E$ loss is based on $S$ and $S^\star$. Finally, we can use $\nabla T$ for motion planning. With only $S^\star$, the problem becomes a trajectory optimization problem, and it is easy to get into the local minimum, while our method can provide an approximate global solution. We have explained this issue in Section 3.2.
>
> **3b.** *What is the motivation of metric learning?*
>
> **Authors:** It is well-received in the learning community that, when a learned function has a certain property, such as being a metric or being Lipschitz, the neural network should adhere to these properties [1]. In our case, our learned value function is exactly a distance metric between start and goal configurations. The three properties, non-negativity, symmetry, and triangle inequality, hold for such functions. By aligning the network output with these properties, we enhance the training efficacy and robustness.
>
> [1] Optimal Goal-Reaching Reinforcement Learning via Quasimetric Learning

---

> ### Author Response · Authors · 2024-11-17
> **For Reviewer 3RM5 (Part 2)**
>
> **3c.** *Why is $L_1$ distance preferred to Euclidean distance?*
>
> **Authors:** To illustrate why our distance metric is preferable, consider a scenario where the environment is a 2D circle. Let’s denote the two poles of the circle as points **A** and **B**. We know that there are two shortest paths (two half-circle arcs) between these points. However, if we use Euclidean distance, there exists only one shortest path between the latent representations of points **A** and **B**. The midpoints of the two half-circle arcs map to the same point in the latent space, yet the actual distance between them on the 2D circle is not zero.
>
> As a result, to accurately reflect the true distance between points **A** and **B**, the 2D circle would need to be compressed into a single straight line. In contrast, by utilizing our distance metric with $L_1$ distance, we can avoid such inconsistencies. For example, if we transform the circle into a 2D rectangle with points **A** and **B** mapped to two opposite corners and the midpoints to the other corners, we can learn the correct distances. We have included the above discussion and its illustration in our paper in Section 4.2.1.
>
> **4.** *What is the benefit of MPC?*
>
> **Authors:** The main advantage of Model Predictive Control (MPC) is its flexibility rather than its speed. In the original work on NTFields, the learned value function gradients were utilized directly for motion planning. In contrast, with MPC, the learned value function serves only as a heuristic to guide the motion planning process. The MPC algorithm employs search-based techniques to calculate feasible and nearly optimal solutions based on these heuristics. This approach allows the online planning algorithm to accommodate additional constraints, such as dynamic obstacles or new user-defined cost functions that can be layered on top of the original value function. Furthermore, our results indicate that combining MPC with our approach leads to an improved computation time of 0.056 seconds, which is even faster than the 0.074 seconds recorded with NTFields. This point is discussed in Section 4.3.
>
> **5.** *Mention known environment, free space.*
>
> **Authors:** We have now explicitly mentioned this assumption in section 3.1.
>
> **6.** *Fig.2 is a key result. However, for people outside the field, it is unclear how to interpret it. It would be great to add some guidance.*
>
> **Authors:** We have revised our description in 5.2 to provide guidance on interpreting the results presented in Fig. 3 (previous Fig. 2).
>
> **7.** *Clarification about the attention mechanism*
>
> **Authors:** Our network architecture utilizes implicit neural representations, which are MLP-like structures that take spatial coordinates as input to predict signals such as the travel time field. To condition this structure on environmental input, we compute a latent code from the environment's point cloud and integrate it with the MLP. While a straightforward approach is to concatenate the latent code with one hidden layer, recent research has shown that using attention mechanisms significantly enhances the network's representational power.
>
> The attention mechanism allows the network to selectively focus on relevant parts of the environment, dynamically adjusting its output based on the spatial context provided by the point cloud. This capability improves the network's ability to generalize across diverse and unseen environments, making it an essential component for robust motion planning in complex settings. We have revised our Section 4.2.2 to include the above contextual information.
>
> **8.** *Connection between online RL and NTFields*
>
> **Authors:** We agree that physics-informed motion planning methods can be viewed as a specific form of offline goal-conditioned RL. These approaches focus on solving motion planning tasks by leveraging the structure of the environment rather than directly interacting with it, as in classical RL.
>
> While traditional RL often involves learning policies under complex dynamics, physics-informed methods simplify dynamics (e.g., assuming constant velocity control) and instead focus on handling more intricate environmental constraints. The Eikonal equation provides a principled way to represent the cost-to-go function in such settings, effectively embedding goal-conditioned behavior directly into the training process.
>
> In Sec. 2, we have highlighted the synergy between our method and offline RL.
>
> **9.** *As $S$ is used for collision avoidance, why do we need $L_N$?*
>
> **Authors:** $L_N$ is used to provide an initial guess for the collision-avoiding behavior of the value function. Our findings indicate that NTFields often get stuck in local minima during the early stages of training, making it difficult for them to escape without the influence of this term.

---

> ### Author Response · Authors · 2024-11-22
> **Looking forward to discussion and reassessment of our work**
>
> Dear Reviewer,
> thank you for your valuable feedback. We have revised the manuscript and provided detailed responses to address your concerns.
> We hope our revisions meet your expectations and kindly request your reassessment.
> Thank you for your time and consideration.

---

> > ### Author Response · Authors · 2024-11-25
> > **Request for discussion or reassessment of our work**
> >
> > Dear Reviewer,
> > Thank you once again for your valuable feedback. This is a gentle reminder that the final date for discussions between reviewers and authors is November 27. If you have any further questions, please let us know. We have revised the manuscript and provided detailed responses to address your concerns. We hope our revisions meet your expectations and kindly request your reassessment. Thank you for your time and consideration.

---

### Author Response · Authors · 2024-11-17
**Summary for all reviewers and area chair**

We would like to extend our gratitude to all reviewers for their detailed and constructive feedback. Below, we provide individual responses to each reviewer's comments, with changes in the paper highlighted in blue for your convenience. In addition to addressing all minor concerns and clarification questions, we would like to emphasize the main modifications as follows:

- We have revised the writing in our paper to offer clearer insights into our TD loss and metric learning.
- We have augmented prior self-supervised methods, NTFields and P-NTFields, with our attention-based environment encoding, incorporating them as baselines in our generalization experiments.
- We provide intuitive details regarding our choice of hyperparameters across different environments.
- We have included additional ablation results and tables presenting both seen and unseen results separately.

Additionally, we would like to highlight that physics-informed self-supervised learning for robot motion planning is a new and emerging research area. Our method demonstrates a significant performance boost compared to previous self-supervised learning-based methods. We believe this new line of research has great potential, as it allows learning-based motion planning to proceed without the need for expert data.  We show in this paper that our method solves the Eikonal equation more accurately and offers better scalability, generalization, and applicability to more challenging planning problems than previous self-supervised learning-based methods.

---

### Meta-Review · Area_Chair_UXAd · 2024-12-26

**Metareview:**

The authors propose a method for finding collision-free robot paths using self-supervised learning. In particular, the authors propose a new discrete-time loss based on temporal difference learning, which solves the Eikonal equation more accurately in order to complete tasks in complex and previously-unseen tasks. They demonstrate on various environments and robots with different numbers of degrees of freedom.

The reviewers found the approach to be novel and interesting, though concerns were raised about clarity -- many things remain unclear.

Strengths:
The method is novel, and they provide an interesting way of computing cost-to-go which might be very useful for robot motion planning in the future. The method is self-supervised, training is fast, and there are sufficient ablation experiments to understand the authors' choices.

Weaknesses:
The authors should better explain the Eikonal equation, and especially how it can be incorporated with other planners to achieve better success rates. Reviewer 7oHd in particular points out that the proposed method has lower than expected success rates - if this is due to the motion planner, the authors ought to incorporate their method into a better planner. Without this, it's very hard to see if there's a real improvement over the state of the art.

In addition, the method proposed is very complex, which on its own may impact its usefulness.

It's also unclear how well the method will generalize.

**Additional Comments On Reviewer Discussion:**

The authors took the reviews into account and improved the quality of the paper. After the authors' detailed responses, the consensus was that the updated manuscript better explained the method (3RM5) and that it was novel and demonstrated improved performance. Reviewer 7oHD did not change their score, and raised some very important concerns which demonstrate important weaknesses of the paper; the authors should attempt to address these in the future, but the paper remains valuable even as it stands now.

---

### Decision · Program_Chairs · 2025-01-22

Accept (Poster)